# Vertical Structure of Ocean Surface Currents Under High Winds from Massive Arrays of Drifters

John Lodise[1], Tamay Özgökmen[1], Annalisa Griffa[2], Maristella Berta[2]

[1]RSMAS, University of Miami, Miami, FL, 33149, USA
[2]CNR-ISMAR, Lerici, Italy

*Correspondence to*: John Lodise (jlodise@rsmas.miami.edu)

**Abstract.** Very near surface ocean currents are dominated by wind and wave forcing and have large impacts on the transport of buoyant materials in the ocean. Surface currents, however, are under resolved in most operational ocean models due to the

difficultly of measuring ocean currents close to, or directly at, the air-sea interface with many modern instrumentations. Here, observations of ocean currents at two depths within the first meter of the surface are made utilizing trajectory data from both drogued and undrogued CARTHE drifters, which have draft depths of 60 cm and 5 cm, respectively. Trajectory data of dense, co-located drogued and undrogued drifters, were collected during the LAgrangian Submesoscale ExpeRiment (LASER) that took place from January to March of 2016 in the Northern Gulf of Mexico. Examination of the drifter data

reveals that the drifter velocities become strongly wind- and wave-driven during periods of high wind, with the pre-existing regional circulation having a smaller, but non-negligible, influence on the total drifter velocities. During these high wind events, we deconstruct the total drifter velocities of each drifter type into their wind- and wave-driven components after subtracting an estimate for the regional circulation, which pre-exists each wind event. In order to capture the regional circulation in the absence of strong wind and wave forcing, a Lagrangian variational method is used to create hourly velocity

field estimates for both drifter types separately, during the hours preceding each high wind event. Synoptic wind and wave output data from the Unified Wave INterface-Coupled Model (UWIN-CM), a fully coupled atmosphere, wave and ocean circulation model, are used for analysis. The wind-driven component of the drifter velocities exhibits a rotation to the right with depth between the velocities measured by undrogued and drogued drifters. We find that the average wind-driven velocity of undrogued drifters (drogued drifters) is ~3.4-6.0 % (~2.3-4.1 %) of the wind speed, and is deflected ~5°-55°

(~30-85°) to the right of the wind, reaching higher deflection angles at higher wind speeds. Results provide new insight to the vertical shear present in wind-driven surface currents under high winds, which have vital implications for any surface transport problem.

## 1. Introduction

Very near surface currents are especially sensitive to wind and wave forcing, which dominate the dynamics in the

upper few centimeters of the ocean (Wu, 1983). Plastics at the surface have been observed to be transported to our coasts by wind- and wave-induced currents and can be transported differentially depending on their buoyancy, which dictates their

positioning in the upper water column (Isobe et al., 2014). Through numerical modeling, Le Hénaff et al. (2012) found that wind- and wave-induced currents had a strong impact on the fate of surface oil during the Deepwater Horizon oil spill in 2010. Another modeling study showed that in extreme events like hurricanes, Stokes drift, or the forward velocity induced by the depth decaying orbital motion of waves (Stokes, 1847), plays a major role in accurately predicting the movement of

Lagrangian particles at the surface (Curcic et al., 2016). Despite the drastic impact on the dispersal of buoyant pollutants, wind- and wave-driven dynamics in the upper few centimeters are poorly understood and not resolved in modern operational ocean models, which have used a wide range of parameterizations of wind-driven dynamics over relatively deep surface layers, forced with climatological winds (Chassignet et al., 2003; Chassignet et al., 2007).

Observational data that captures the vertical shear within the first meter of wind-driven surface currents is very

limited in the real ocean as well. This is mainly due to the limitations of the instrumentation used to gather this physical data. The majority of Lagrangian drifter studies have used classic drifter designs (e.g. CODE (Davis, 1985) and SVP drifters (Lumpkin and Pazos, 2007)) that span depths ranging from 1–30 m (Lumpkin et al., 2017), and are therefore unrepresentative of the wind forced current at the very surface. Santala and Terray (1992) describe the difficulty in measuring surface velocities, as well as near surface shear, due to wave bias present when utilizing any instrumentation

whose motion is dependent on wave action. Acoustic Doppler Current Profilers (ADCP's) are not able to accurately measure velocities near an undulating boundary, making it difficult to sample surface layers above depths of 0.5 m (Cole and Symonds, 2015; Sentchev et al., 2017). High Frequency (HF) radars, excluding those with multi-frequency capabilities, are known to measure vertically integrated surface currents, with about 80 % of the radar signal originating in the upper 1 m, depending on the electromagnetic frequency ($\sim$ 16 MHz) of the radar, such that vertical shear within this depth is not

detectible in the measurement (Stewart and Joy, 1974; Röhrs et al., 2015). Some work, however, has shown promise in detecting vertical shear by calculating Doppler shifts from multiple Bragg wave peaks simultaneously (Ivonin et al., 2004).

Using a novelty suite of instruments in the Gulf of Mexico, Laxague et al. (2017) observed wind- and wave-induced surface currents within 1 cm of the surface, under low winds (U10 = 4 m s$^{-1}$), to be twice as fast as the average current over the first 1 m and four times as fast as the average current over the upper 10 m (the upper 10 m average being equal to $\sim$0.16

m s$^{-1}$). Another study using drifters of multiple draft depths and HF radar data led to a similar result, showing that HF radar velocity measurements taken at $\sim$2-3 m depth, were on average, only 55.7 % of that measured with ultra-thin drifters with 5cm draft depths, whose absolute velocity was on the order of 0.3 m s$^{-1}$ (Morey et al., 2018). Comparison of surface velocities measured with ultra-thin drifters of 5 cm and 10 cm drafts depths also showed a rapid decay of velocity away from the surface, with the 10 cm drifters traveling 10 % slower on average (Morey et al., 2018). The sharp change in wind- and

wave- induced velocities observed in these two studies reveal the need for increased vertical resolution of these surface measurements (Laxague et al., 2017; Morey et al., 2018).

Classical Ekman theory states that wind-driven currents at the very surface travel at a deflection angle of 45° to the right of the wind in the northern hemisphere, when one assumes a balance between Coriolis and friction under stationary,

homogeneous conditions (Ekman, 1905). Laboratory studies have shown that total surface drift currents induced by wind and waves combined, travel at ~ 3.1 % of the wind velocity, with the wind-induced drift decreasing and the wave-induced drift increasing, with increasing fetch (Wu, 1983). Various observational studies on wind-driven currents over the upper ~1 m of the surface, which utilized a range of different instruments including HF radars, ADCPs and drifters of various types ( i.e.

CODE, iSphere or undrogued SVP drifters) have reported a wide range of deflection angles ranging from 15° to 90° to the right of the wind at varying wind speeds (Ardhuin et al., 2009; Poulain et al., 2009; Kim et al., 2010; Röhrs and Christensen, 2015; Sentchev et al., 2017; Berta et al., 2018). Individual results and details of these studies are presented in Table 1. Combined results of these studies produce a range of estimates for the magnitude of wind-induced currents, with reported values ranging from 0.4 % to 5 % of the wind velocity, over the upper ~1 m, showing that results may vary significantly

based on environmental conditions and methodology (Ardhuin et al., 2009; Poulain et al., 2009; Kim et al., 2010; Röhrs and Christensen, 2015; Sentchev et al., 2017; Berta et al., 2018). Changes in upper ocean stratification and mixed layer depth have been shown to have large effects on the deflection angle and relative velocity of surface currents with respect to the wind (Kudryavtsev and Soloviev, 1990; Sutherland et al., 2016). Rascle and Ardhuin (2009) showed how the complicated relationship between wave-induced mixing and varying stratification can result in quasi-Eulerian wind-driven surface

currents ranging from ~1 to 3 % of the wind speed, having deflection angles ranging from ~35°-90°.

      Here, we use drogued and undrogued CARTHE drifters, having draft depths of 60 cm and 5 cm, respectively (Novelli et al., 2017), in order to observe very near surface currents in the Northern Gulf of Mexico during high wind conditions. The data used for analysis were collected during the LAgrangian Submesoscale ExpeRiment (LASER), a campaign in which over 1000 biodegradable CARTHE drifters were deployed during the winter of 2016 (Haza et al., 2018).

Utilizing a subset of the CARTHE drifters from the experiment, we focus on three synoptic scale, high wind events during which, wind and waves seem to dominate the forcing of the surface flows measured by both drifter types. During these high wind events, we deconstruct the full drifter velocities of each drifter type into their wind- and wave-driven components after subtracting an estimate for the regional circulation which pre-exists each wind event. We then report on the vertical shear of the wind-driven component calculated from the undrogued and drogued drifter velocities. To the authors' knowledge these

findings are the first reported estimates of vertical shear in upper 1 m, involving a direct velocity deconstruction and differentiation between wind- and wave-driven components under high wind conditions (~12-20 m s$^{-1}$).

      Upon inspection of the total velocity of the drifters during the high wind events, we hypothesize that there are three dominant components that drive the drifter velocities: A wind-driven component, a wave-driven component, and the regional circulation that pre-existed each high wind event. Assuming a simple linear superposition of the velocity components, we

define the total velocity of each drifter type as the depth averaged integral:

$$\bar{u}_T = \frac{1}{h} \int_0^h [\, u_s(z) + u_w(z) + u_{rc}(z) \,] dz, \tag{1}$$

where $u_w$ is the purely wind-driven velocity, $u_s$ is the Stokes drift velocity, $u_{rc}$ is the regional circulation that exists before the increase in synoptic winds, z is the depth being evaluated and h is the draft depth of the respective drifter. Given this definition of the total drifter velocities, we neglect possible nonlinear interactions between velocity components, which will be considered in future investigations.

The three components in Eq. (1) were estimated as followed: $u_s$ was calculated numerically with the Unified Wave INterface-Coupled Model (UWIN-CM, Chen et al., 2013) during the LASER campaign and stored for later analysis, $u_{rc}$ is estimated from the dense population of drifters in the region in the hours preceding substantial increases in synoptic winds using the LAgrangian Variational Analysis (LAVA, Taillandier et al., 2006) to create Eulerian velocity fields of the regional circulation, and we solve for $u_w$ by subtracting the estimated regional circulation and Stokes drift velocity from the total velocity of the drifters. To evaluate the performance of the LAVA-derived velocity fields, we use AVISO Sea Surface Temperature (SST) and absolute geostrophic velocity products, in addition to ocean current velocity fields from the Navy Coordinate Ocean Model (NCOM) for comparison.

In the Northwestern Mediterranean Sea, the underlying geostrophic currents were shown to retain their structure and influence the surface flow under high wind forcing for time periods on the order of 2-3 days (Berta et al., 2018). In the present study, the timescales over which the flow features of the regional circulation, $u_{rc}$, retain their structure under strong winds is difficult to determine given the current dataset. However, the subtraction of the pre-existing regional circulation from the total velocity of the drifters during the high wind events, shows a-posteriori, that the pre-existing regional circulation retains its structure, to a reasonable extent, during the high wind analysis periods on which we focus. After subtracting the pre-existing regional circulation and numerically calculated Stokes drift velocity, $u_s$, from the full velocity of the drifters, $\bar{u}_T$, during each period of high wind, we are left with an estimate for the average, purely wind-driven component, $u_w$, of each drifter type.

The paper is organized as follows: Section 2 describes the CARTHE drifters, the configuration of UWIN-CM and NCOM, and the AVISO data products. Section 3 explains the LAgrangian Variational Analysis (LAVA) used to create the estimated Eulerian velocity fields of the pre-existing regional circulation, as well as the calculations involved in the deconstruction of the total drifter velocities, $\bar{u}_T$, along each drifter trajectory during the high wind events. Results are presented in section 4, with a discussion of the results following in section 5. Concluding remarks are presented in section 6.

## 2. Data

### 2.1 CARTHE Drifter

The CARTHE drifter is a biodegradable surface drifter that consists of a Spot Trace GPS unit by Global Star, a torus float which contains the GPS housing, two interlocking panels that form the drogue, and a flexible rubber tube that connects the drogue and float (Novelli et al., 2017). During the LASER experiment, over 1000 CARTHE drifters were deployed with drogues, however, over the first 7 weeks of the experiment approximately 40 % of the drifters lost their drogues (Haza et al.,

2018). Drogued and undrogued drifters have draft depths of 60 cm and 5 cm, respectively, and have been extensively analyzed with respect to their specific drift characteristics during laboratory experiments performed by Novelli et al. (2017). For this study, a subset of co-located drogued and undrogued drifters from the LASER campaign are used for analysis, due to their opportunistic location during the passage of large atmospheric fronts across the Northern Gulf of Mexico.

Drogue loss during the experiment mostly coincided with large storm and wave events and the precision of the determined time of drogue loss was 0.5 to 3 hours for 85 % of the drifters (Haza et al., 2018). The method for drogue loss detection is based on the differential velocities of the drifters, as the undrogued drifters are preferentially accelerated by the higher velocities of wind- and wave-driven currents present in the shallower surface layer in which the undrogued drifters reside. In addition, undrogued drifters display a decreased, more sporadic GPS transmission rate, due to their tendency to be

flipped by large or breaking waves, which points the GPS antennae downwards, and reduces the ability of the GPS to transmit until the drifter is flipped upright again. Despite this, 80 % of the time intervals between transmissions by undrogued drifters are still between 4.5 and 5.5 minutes, but with notably higher outliers than the drogued drifters (Haza et al., 2018). The algorithm for drogue detection used by Haza et al. (2018) was validated using a subset of 50 drifters with known drogue status and was shown to distinguish drogued and undrogued drifters with an accuracy of 94-100 %. This very

successful drogue detection algorithm has provided the opportunity to utilize both drogued and undrogued drifters to study the variation of very near surface currents with depth (Haza et al., 2018).

       GPS transmissions reported each drifter's location every 5 minutes during the experiment, with an accuracy of about 7 m. In addition to the extensive categorization of drogued and undrogued drifters, the drifter trajectory data were also quality controlled for missing transmissions and linearly interpolated to regular 15-minute intervals (Haza et al., 2018). Velocities

were then calculated, resulting in estimates for the average velocity of each drifter over 15-minute intervals.

       When using drifters to study ocean currents, concerns about different sources of velocity slip must be addressed. Extensive laboratory testing performed by Novelli et al. (2017) using half-scale drogued and undrogued drifters showed that in the absence of wind and waves drogued (undrogued) drifters travel within 0.01 m s$^{-1}$ (0.02 m s$^{-1}$) of the mean Eulerian current averaged over the draft depth of the given drifter. Under the effects of waves, observed in the absence of wind, the

undrogued drifter unsurprisingly feels an acceleration due to stokes drift, but the drogued drifter feels a reduced wave-induced acceleration due to the flexible tether holding together its float and drogue. This mechanical decoupling partially removes the effect of the Stokes drift acting on the drogued drifter, mainly dampening the effects of stokes drift above the drogue (Novelli et al., 2017).

       To characterize the slip velocity associated with wind, waves, and Eulerian current, Novelli et al. (2017) defines the

"absolute" slip velocity as the difference between the velocity of a drifter and the depth integrated current over each drifters' draft. The absolute slip velocity of both drogued and undrogued drifters during laboratory testing was found to decrease with increasing wind speed, decreasing from 3 cm s$^{-1}$ to 0.1 cm s$^{-1}$ for the drogued drifter and from 14 cm s$^{-1}$ to 1 cm s$^{-1}$ for the undrogued drifter, for wind speeds from 8.1 m s$^{-1}$ to 23 m s$^{-1}$. This phenomenon is thought to be caused by wind separation

from the ocean surface due to the presence of surface gravity waves (Novelli et al., 2017). Another laboratory experiment, focused on measuring the turbulent air flow over wind generated waves in a similar wind-wave tank, found wind separation to occur over 90 % of short wind waves at 10 m wind speeds of ~16 m s[-1] (Buckley and Veron, 2017). The extent to which wind separation occurs and its effectiveness of sheltering the drifters from wind-slip in the real ocean is difficult to quantify

given the scale differences between the laboratory wave tank and open ocean. However, another recent experiment using full size CARTHE drogued and undrogued drifters in the real ocean, alongside a suite of instruments including an ADCP and polarimetric camera, showed that velocities calculated using both drifter types fell within the range of velocities measured by other instrumentation over the corresponding draft of each drifter (Laxague et al., 2017). Because the extent to which velocity slip affects the drifters in the real ocean is not well known during high wind and wave conditions, we choose not to

attempt a correction of the drifter velocities to account for such measurement errors.

**2.2 UWIN-CM**

During the extent of the LASER campaign, daily 72 hour real-time forecasts were produced using the Unified Wave Interface-Coupled Model (UWIN-CM), running a fully-coupled atmosphere, surface gravity wave, and ocean circulation system (Chen et al., 2013; Chen and Curcic, 2016). The UWIN-CM model atmospheric component is comprised of the

atmospheric non-hydrostatic Weather Research and Forecasting (WRF) model featuring Advanced Research WRF dynamical core with 4 km horizontal resolution over the Gulf of Mexico with 36 vertical layers (Skamarock et al., 2008; Haza et al., 2018).

The surface gravity wave model is the University of Miami Wave Model v2 (UMWM, Donelan et al., 2012), with the same 4 km resolution as the atmospheric component. The three-dimensional Stokes drift velocity fields are calculated as

the full integral:

$$u_S = \int\limits_{0}^{2\pi} \int\limits_{k_{min}}^{k_{max}} \omega k^2 \frac{\cosh[2k(d+z)]}{2\sinh^2 kd} F(k,\theta) dk d\theta, \tag{2}$$

where $\omega$ is the angular frequency, k is the wavenumber, d is the mean water depth, z is the depth being evaluated, F is the wavenumber energy spectrum, and $\theta$ is the direction of the waves (Stokes, 1847; Phillips, 1977). Over the entirety of LASER, k ranged from 0.0039 rad m[-1] to 16.0739 rad m[-1], in 37 logarithmic increments, which corresponds to a range of

wavelengths from 1611 m to 0.39 m, respectively. The Hybrid Coordinate Ocean Model (HYCOM) v2.2 is used as the ocean circulation model, however no model output from this circulation model is used for the analysis of wind-driven currents. The surface layer of the HYCOM model has a minimum thickness of 3 m, which makes the model unrepresentative of the depths sampled by the drifters and very difficult to validate with observational data (Wallcraft et al., 2009). Comparison of drifter data and surface velocities from the circulation model showed obvious discrepancies at the spatial scales necessary for this

study.

The coupling between the model components is as follows: WRF passes radiative and heat fluxes, as well as precipitation rates to the ocean model, as well as the air density and wind profiles to the wave model. The wave model (UMWM) passes vectorial atmosphere stress and vectorial ocean stress to the atmosphere model and ocean circulation model, respectively. The ocean circulation model passes SST to the atmosphere model and surface current fields and ocean density to the wave model. WRF and UMWM fields are exchanged every 60 s, while HYCOM fields are exchanged every 120 s (Chen and Curcic, 2016).

The UWIN-CM is initialized daily using initial and boundary conditions from Global Forecast System (GFS) and global HYCOM fields along with the previous day's UMWM wave forecast. Coupling between models is executed using the Earth System Modeling Framework (ESMF) in which all components are exchanged between models every minute (Hill et al. 2004). Hourly 10 m wind (U10) and Stokes drift velocity data from the $24^{th} – 48^{th}$ hour of each daily 72-hour forecast were stored and used for the analysis performed in this study. Initially having a temporal resolution of 1 hour, the UWIN-CM model output was interpolated to 15-minute intervals to match the time resolution of the drifter data.

### 2.3 NCOM

The ocean circulation component of NCOM is chosen for comparison to the LAVA-derived velocity fields, over the ocean component of the UWIN-CM, due to its increased resolution of 1 km and data assimilation of available satellite and in-situ observations (Jacobs et al., 2014). NCOM forecasts were produced in real-time during the LASER campaign, and were utilized to guide scientists in the field. The NCOM velocity fields are produced on 3-hour intervals, having a vertical resolution of 2 m from 0-10 m, with larger vertical layers as depth increases. The surface wind stress for the NCOM ocean circulation are calculated using atmospheric conditions from the Coupled Ocean/Atmosphere Mesoscale Predictions System (COAMPS, Hodur, 1997). Surface heat fluxes are determined from ocean model SST and 10-m air temperature and humidity, through bulk flux formulations. Locally generated tidal forcings are implemented using the Oregon State University global Ocean Inverse Solution (OTIS) (Egbert and Erofeeva, 2002). During LASER, the NCOM forecasts were able to accurately represent the mesoscale velocity field in the Gulf of Mexico, to a certain extent, due to the assimilation of altimetry-based SSH (Haza et al., 2019). However, Haza et al. (2019) found the wind stress parameterization implemented with COAMPS to considerably underestimate the wind-driven component of the surface current when trying to reconstruct drifter trajectories using NCOM surface velocities. For this reason, the authors choose not to rely on the NCOM wind stress for information on wind-driven drifter velocities.

### 2.4 AVISO Data

Absolute geostrophic currents and SST products, produced by Collecte Localisation Satellites (CLS) from AVISO satellite altimetry data, are used in this study to assess the performance of the estimated velocity fields created using the drifter data and LAVA in the hours preceding each high wind event. The absolute geostrophic current, is deduced from fields of absolute dynamic topography using stencil width methodology, which provides daily (24 hour intervals) velocity fields

with horizontal resolutions of 0.25° (~28 km) (Arbic et al., 2012). The AVISO SST products in the Gulf of Mexico are measured by 4 intercalibrated satellite infrared radiometers and have a resolution of 0.02° (~2 km) in latitude and longitude. Each 24-hour period, over which the SST measurements are processed and averaged, is centered on the 0th hour of each day.

## 3. Methods

### 3.1 High Wind Events and Region of Interest

The domain for this study lies to the east and southeast of the Mississippi River Delta spanning from 27.5°N to 30.5°N and 90°W to 86.5°W. The spatial extent of the data used for each high wind event are outlined in Fig. 1, along with NDBC buoy locations used for verification of the UWIN-CM wind data. Initial deployment of these drifters occurred during January and February of 2016 with the intent of capturing submesoscale dynamics on spatial scales of tens of meters to tens of kilometers (Haza et al., 2018). The large number of drifters deployed during the LASER campaign, and their relatively long transmission period of about 3 months, provided opportunity to collect data over a range of scales and environmental conditions. The region and time periods chosen exhibit large numbers of co-located drogued and undrogued drifters during the passage of synoptic atmospheric fronts, which drive a large momentum input into the oceanic boundary layer.

In order to characterize the vertical structure of flow features typical in this region, a moving vessel profiler (MVP) collected CTD measurements when possible during the first month of the experiment, focusing on the fronts and eddies that form at the intersection of cold, fresh Mississippi outflow water and the warmer, saltier Gulf of Mexico waters. Figure 2 shows a typical transect (outlined in Fig. 1) of potential density, salinity, and temperature across a frontal structure in the region. These transects display less dense Mississippi outflow waters extending down to ~30 m depth from the surface, flowing over the denser interior gulf water. The interior gulf water, seen to extend to the surface towards the eastern end of the transects, shows a well-mixed surface layer down to ~80 m depth. Other transects across frontal areas measured with the MVP show very similar structures in this region, making Fig. 2 representative of the typical stratification observed between these water masses.

For this study, we choose to focus on three high wind events that occurred on January 22$^{nd}$, February 24$^{th}$ and March 20$^{th}$ of 2016. Available wind data from nearby NDBC buoys (BURL1, 42040 and 42012), along with the UWIN-CM wind data associated with each drifter position during and before the high wind event are plotted in Fig. 3. The time periods over which we perform the deconstruction of the total measured surface currents during each high wind event are marked by vertical black lines, and the red solid and dashed vertical lines show the start of the hour over which the pre-existing regional circulation is estimated from drogued and undrogued drifters, respectively (Fig 3). Validation of the UWIN-CM U10 output with available NDBC buoy data revealed that during the passage of atmospheric fronts through the domain there exists a fair amount of variability in the modeled wind magnitude resulting, at times, in a large difference between observed and modeled data. For this reason, we exclude any data associated with modeled wind magnitude ranges above 6 m s$^{-1}$ at any given 15-minute time step.

Wave data from the UWIN-CM is also compared to the available NDBC wave data in Fig. 4. Only data from NDBC buoy 42040 during the January high wind event is shown for wave model validation, as there was no wave data collected from buoy BURL1, and buoy 42040 stopped functioning before the February event. Comparison of model output and observational data shows that the significant wave height is modeled accurately, especially under high winds, while the wave period could be slightly underestimated by the wave model. Mean wave direction is initially offset by ~30° at the beginning of the high wind analysis period, but comes into good agreement about halfway through the analysis window. Wave data from buoy 42012 also showed good agreement to the UWIN-CM data near the buoy coordinates during the March event, but as drifters move further offshore during the high wind event, the nearshore location of the buoy becomes less representative of the conditions influencing the drifters.

For further validation of the Stokes drift velocity computed with the UWIN-CM model, we compare higher frequency wave data from buoy 42040 to that of the model. Figure 5a,b shows the 2-D wave energy spectrum from 1 hour during the high wind analysis period from the model output and calculated from the NDBC buoy using the Longuet-Higgins (1963) Fourier expansion method coefficients provided by NDBC. Figure 5c shows the 1-D wave density spectrum of the modeled and observed waves, over the same hour displayed in Fig. 5a,b. For validation of the Stokes velocity used by UWIN-CM, Eq. 2 is used to calculate the Stokes velocity using the 2-D wave spectrum from buoy 42040. Buoy 42040 is a 2.1 m diameter, SCOOP type buoy, that extents down ~ 1.68 m from the water line, and reports wave spectrum data from 0.02 to 0.48 Hz (Bouchard et al., 2018). In order to make a fair comparison, we calculate the Stokes velocity using the 2-D modeled wave spectrum from 0.0313 to 0.5 Hz, which are the closest corresponding frequency bins to that of the NDBC data. We then integrate the modeled Stokes velocity from the surface down to 1.68 m in order to accurately compare the buoy and wave model calculated Stokes velocities. Both the Stokes drift magnitude and direction calculated from the modeled and buoy wave spectrums are plotted in Fig. 5d.

The modeled wave spectrum, both 1-D and 2-D, compare reasonably well to that of buoy 42040, with the obvious difference being the magnitude and slight offset of the peak seen in the spectrum at ~ 0.12 Hz (Fig. 5a,b,c). The model seems to underestimate this peak, while overestimating the contribution from waves in the frequency range of ~0.15 – 0.3 Hz. Figure 5a,b,c displays typical examples of both the modeled and observed wave spectrum during the January high wind analysis period, which results in the slight over estimation of the Stokes drift magnitude calculated from the modeled wave spectrum seen in Fig. 5d. The NDBC buoy could have some trouble accurately recording smaller scale motions from higher frequency waves under intense winds and coinciding strong surface currents, causing some discrepancy. The model and buoy calculated Stokes velocities do show good agreement with regards to direction.

## 3.2 LAgrangian Variational Analysis (LAVA) and Pre-existing Regional Circulation

Initial inspection of the total drifter velocities during each high wind event showed larger than expected spatial variation in drifter velocities, including velocities which depict surface currents traveling to the left of the wind, which challenges previous wind-driven surface current theory and observations. Previous works that have studied instantaneous wind-driven

dynamics over a range of spatial scales have illustrated the need to account for the circulation present before the observed increase in synoptic winds in order to isolate the wind-driven component of the flow (Sentchev et al., 2017; Berta et al., 2018). Based on these studies, we hypothesize that the spatial variability observed in the total velocities is due to the regional circulation that pre-existed the period of increasing winds, which retains its structure on times scales long enough to influence the surface flow during each high wind event.

In order to resolve surface currents in our region of interest at adequate spatial and temporal resolution we utilize the Lagrangian Variational Analysis (LAVA, Taillandier et al., 2006) and the available drifter data in the region to estimate the regional circulation. In a previous study, geostrophic velocities derived from altimetry data and surface Ekman current velocities parameterized using forecasted winds have been shown to have inadequate spatial and temporal resolution, in order to accurately estimate surface currents in the Northern Gulf of Mexico on scales less than the order of 100 km in space and 1 week in time (Berta et al., 2015). LAVA allows us to use the undrogued and drogued drifter trajectory data in the region, to create Eulerian velocity fields that are statistically robust and on the spatial and temporal scales on the order of 10 km and hours, respectively (Taillandier et al., 2006; Berta et al., 2015). LAVA has been used in previous studies to create velocity fields using purely Lagrangian data, as well as blending drifter trajectory data with Eulerian velocity fields derived from altimetry and HF radars. LAVA has proven especially useful when providing Near Real Time (NRT) information that can be useful to first responders of oil spills, search and rescue efforts and other surface transport problems (Taillandier et al., 2006; Chang et al., 2011; Berta et al., 2014; Berta et al., 2015).

Given our hypothesis that during the high wind events, the total velocity of the drifters is partly composed of the regional circulation that pre-existed the onset of high winds, an estimate for this circulation must be removed from the total velocity of the drifters in order to isolate the wind- and wave-driven components. We utilize LAVA and the available drogued and undrogued drifter data in the region to create semi-simultaneous hourly Eulerian velocity fields for both drifter types, separately. The drifters used for the velocity field construction for each event are plotted in Fig. 6. The January 22[nd] event occurred very close to the initial time of deployment while the drogued drifters were still tightly packed spatially and little drogue loss has occurred, leaving a smaller, but adequate, number of undrogued drifters to utilize for the estimate of the pre-existing circulation. The events on February 24[th] and March 20[th], occur long enough after deployment, thus resulting in a more even amount of drogued and undrogued drifters that are also more evenly spread through the region, however drogued drifters do show a tendency to converge upon one another, evident in the spatial organization of the drifters seen in Fig. 6.

LAVA utilizes drifter trajectories in order to create average velocity fields over multiple time steps depending on the parameters R, $T_a$, $\Delta t$ and $\Delta x$, which need to be prescribed before implementation of the analysis. R determines the spatial range over which the velocity of each drifter is spread from the drifter location through finite iterations of the diffusion equation and should be chosen as the typical length scale, or Rossby radius, for the targeted oceanic dynamics to be resolved in the given region. For this analysis, R is set to 10 km, which is same value used for a previous study which utilized drifter data in the same region of the Northern Gulf of Mexico (Berta et al., 2015). $\Delta t$ for this study is 15 minutes, which is the time resolution of the quality controlled drifter trajectories which have been linearly interpolated to regular 15-minute intervals

(Haza et al., 2018). The analysis time scale, $T_a$ is the larger time window over which consecutive velocity fields created using LAVA are averaged and should be shorter than the Lagrangian time scale (Taillandier et al., 2006). Lagrangian time scales within this region have been calculated to be as small as ~1-3 hours, which corresponded to spatial scales ranging from .4-3.5 km (Gonçalves et al., 2019), and as large as ~1-3 days corresponding to spatial scales ranging from ~10-35 km (Ohlmann and Niiler, 2005). Here, $T_a$ is set to 1 hour, which is adequately short given our assignment of 10 km as the typical horizontal length scale being resolved by LAVA. $\Delta x$ is the spatial resolution of the discretized velocity field and needs to be assigned such that $\Delta x < R$, defined here as $\Delta x = 1.5$ km. (Taillandier et al., 2006; Berta et al., 2015). Drifter trajectories within $2\Delta x$ of one another are averaged to become single drifter trajectories positioned along their center of mass before the production of Eulerian velocity fields (Berta et al., 2015).

After the Eulerian velocity fields are created, a kinetic energy mask is also implemented to exclude small velocities that are an artifact of assigning radially decreasing velocities away from drifter locations using LAVA. To avoid these unrealistically small velocities, any values in the Eulerian velocity fields that represent less than 10 % of the hourly averaged kinetic energy in the velocity field are discarded. This relatively low threshold of kinetic energy was chosen to maximize the coverage between the Eulerian velocity field estimates of the pre-existing regional circulation and drifter trajectory data during each high wind event.

The hourly velocity fields are created during low wind conditions, hours before the passage of synoptic scale atmospheric fronts, which produce very high winds over the entire region. To avoid unnecessarily smoothing the Eulerian velocity fields through averaging in time, we choose one hourly velocity field per event, per drifter type, to use for analysis. Hourly Eulerian velocity fields are chosen based on the criteria that there are relatively low wind velocities present, the time gap between the velocity field and onset of high wind isn't needlessly long and how well the Eulerian velocity field overlies the drifter trajectories during the high wind event. The Eulerian velocity fields used for analysis are plotted in Figs. 7 and 8. Each hour over which these velocity fields are created are also indicated in Fig. 3. The Eulerian velocity fields are plotted on top of AVISO derived, 24-hour averages of SST temperature, in order to validate the flow structures that are seen in the reconstructed velocity fields using LAVA. For each high wind event, the drogued and undrogued velocity fields are directly compared to two SST fields. One corresponding as closely as possible in time to the high wind event and one corresponding to the following day, in order to visualize the change in the pre-existing regional circulation during each high wind event (Figs. 7, 8).

Stokes drift estimates from the UWIN-CM during the time periods shown in Figs. 7 and 8, are on average an order of magnitude less than the total drifter velocities for both drifter types, having average magnitudes of 0.02 m s$^{-1}$ for the drogued case and 0.04 m s$^{-1}$ for the undrogued case. Smoothing of the constructed velocity fields that arises from the velocity spreading and averaging performed by LAVA over the length scale R, makes the uncertainties from the influence of Stokes drift insignificant during these low wind conditions. Inevitably there also exists some influence from the wind during these time periods, which is most likely greater than the magnitude of the Stokes drift. However, the qualitative agreement of the

velocity field estimates and the SST fields, evident in Figs. 7 and 8, suggest that both drifter types are capturing the dynamics of the underlying circulation rather than the instantaneous wind-driven effects during these time.

To provide a comparison to the LAVA-derived velocity fields, geostrophic velocity fields from altimetry data and NCOM velocities fields during each high wind event are shown in Fig. 9, plotted on top of associated SST temperature data. During the January and February wind events we select the NCOM ocean velocity at 60 m depth, at the beginning of the high wind analysis periods shown in Fig. 3. This depth is chosen based on the stratification shown in Fig. 2, 60 m being just above the sharp pycnocline seen at ~80m, thus being able to capture the mixed layer dynamics driving the regional flow at the surface, but deep enough as to not be affected by any wind stress in the model. As for the March event, much of the domain is bathymetrically limited due to its proximity to the coast, forcing the authors to use a shallower depth of 15 m for comparison. Velocity fields during, rather than before, each high wind event are used to allow the model to account for any structural changes the flow may have experienced during the time gap between the LAVA-derived velocity fields and analysis period during high winds. Such evolution of the regional circulation will not be observed by the geostrophic velocities from altimeter data as spatial and temporal resolution is much coarser.

### 3.3 Deconstruction of Total Surface Current

The total drifter velocities, $\bar{u}_T$, calculated between each 15-minute drifter position during the high wind events, is stored at the later drifter coordinate used for each calculation. The time gap between the observation of the regional circulation, $u_{rc}$, and the beginning of the high wind analysis period, is chosen such that the increase in winds shows a noticeable change in all the drifter trajectories across the given domain. Investigation of wind-driven currents at high temporal resolution has shown that the surface current response time to changing winds can be described as quasi-instantaneous, with the surface currents lagging the wind by ~40 minutes (Sentchev et al., 2017). Having a temporal gap between these observations ensures the surface velocities, across the region, have ample time to adjust to the increasing winds before any calculations are made. In some cases, the time gap is extended to allow drifters to enter the region where the corresponding observation of the pre-existing regional circulation exists. In addition, we stop the analysis for each high wind event at the apex of the increasing wind velocity magnitude, beyond which the drifters exhibit large inertial motions due to the decrease in momentum input from the wind. The drifter trajectories, during the period over which the deconstruction of the total drifter velocities are made, are shown in Fig. 10.

From the full drifter velocities, calculated on 15-minute intervals during each wind event, we subtract the nearest point velocity in the estimated LAVA velocity field of the pre-existing regional circulation, $u_{rc}$. Each velocity field has a spatial resolution of 1.5 km, which is adequate to preserve the flow features given the typical length scale of 10 km in the region (Berta et al., 2015). In the event that the nearest point to the drifter location in the LAVA velocity field estimate does not have a value, due to a lack data in the velocity field, the data is discarded. The trajectories in Fig. 10 are color coded to show where the drifter trajectories and Eulerian velocity fields overlap during the high wind event and where data coverage is missing. Subtraction of the regional circulation estimates from altimeter based geostrophic velocities and NCOM velocity

fields (Fig. 9) are also performed in the same manner. For accurate comparison, any data points along the drifter trajectories that were excluded during the subtraction of the LAVA velocity fields estimates, were also excluded from the calculation involving the AVISO and NCOM velocity fields.

The subtraction of multiple hourly adjacent LAVA velocity fields, earlier and later in time, were examined a-posteriori for each event and found to produce similar end results. The hourly LAVA velocity fields chosen for this analysis were those that had the most data points in common with the respective drifter trajectories during the high wind event, or those that retained a similar amount of data points during the subtraction (within 3.5 % of the maximum amount of retained trajectory data for each specific case) but resulted in larger decreases in the standard deviation of the deflection angle between drifters and wind velocity. This decision is motivated by the assumption that the isolated effects of wind and waves will result in a more directionally uniform velocity field, than that of a surface flow being influenced by the regional circulation estimates shown in Figs 7 and 8. Thus, by removing more variation in the deflection angles between the wind and drifter velocities, we better represent the combined wind- and wave-driven components, assuming the wind and wave directions are well aligned. All velocity estimates and calculations are performed using drogued and undrogued drifter data, separately.

After subtracting the estimates for the pre-existing regional circulation from the full drifter velocities during the high wind events, the remaining velocity is an estimate for the combined wind- and wave-driven flow ($u_s + u_w$), referred to here after as the wind-wave-driven velocity. To further deconstruct this velocity estimate, we subtract the UWIN-CM modeled Stokes drift velocity ($u_s$) fields, again at the nearest point to each drifter location for each 15-minute interval. Because the large wind-driven events being analyzed are of synoptic scale, the UWIN-CM modeled Stokes velocity and U10 winds are relatively uniform in the region of study, making the 4 km resolution of the data adequate to perform meaningful calculations in this manner.

Because the drogued drifters display a filtering characteristic of the surface Stokes drift, as illustrated in laboratory testing performed by Novelli et al. (2017), the stokes drift at 0.4 m depth is used for the calculation with drogued drifter-derived velocities (0.4 m being the vertical center of the drogue). From the undrogued drifter associated velocities, we subtract the surface ($z = 0$ m) Stokes drift velocity. After subtracting the wave-driven component, $u_s$, from the wind-wave-driven velocity, we are left with an estimate for the average wind-driven drifter velocities, $u_w$, over each respective drifter draft depth. The average wind-wave-driven and wind-driven drifter velocities are compared to the modeled U10 wind data, using the U10 wind velocity datum at the nearest point to each drifter location at every time step.

### 4. Results

The main flow features seen in the estimated Eulerian velocity fields and SST fields show evidence of flow features having spatial scales on the order of tens of kilometers (Figs. 7 and 8). Observations of smaller flow features in the velocity fields are limited by the 1.5 km resolution and typical length scale of 10 km set by the chosen LAVA configuration. The regional circulation observed, by either drifter type, prior to the wind events on February 24[th] and March 20[th] show an

abundance of meanders, eddies and frontal features, whereas the regional circulation pre-existing the wind event on January 22nd suggests that a somewhat larger feature, closer to the order of 100 km, is driving the majority of the flow to the northeast with smaller scale variability seen throughout the flow. The velocity field vectors for all panels in Figs 7 and 8, show good agreement to the structures highlighted by the SST data, supporting the validity of the LAVA estimated fields.

Features on these spatial scales (10-100 km) are known to have characteristic time scales on the order of days to weeks (Özgökmen et al., 2016) which suggests that the main flow features observed in the pre-existing regional circulation, are likely retain their overall structure, to a certain extent, during of the high wind analysis periods presented here. This is also supported by the SST fields averaged over the 24-hour period following each high wind event shown in the right-side panels of Figs. 7 and 8. Although the SST field change somewhat during the high wind event, the pre-existing circulation estimates seem to be valid for an appreciable amount of time, evident in the overlap between eddies, fronts, and filaments observed in the SST field and the velocity vectors created with LAVA 2-3 days previously.

Comparison between Figs. 7 and 8 with Fig. 9, displays robust discrepancies between the velocity fields created using LAVA and drifter data, altimetry data, and NCOM. The coarse resolution of the geostrophic velocities does not capture the circulation highlighted in the SST data. The geostrophic velocities are also appreciably smaller than either LAVA or NCOM (vectors for the geostrophic velocities in Fig. 9 are magnified by a factor of 2 compared to those of the NCOM circulation). Velocities showing the NCOM circulation, seem to better compliment the structures observed in the SST data compared to the geostrophic velocities (Fig. 9), however compared to the LAVA-derived velocity fields in Figs. 7 and 8, there still exists a fair amount of disagreement between the NCOM and LAVA velocity fields, possibly the most pronounced example being the velocity fields associated with the January wind event.

The trajectories of both drogued and undrogued drifters during the hours of increasing winds, over which the total velocity deconstructions were performed are shown in Fig. 10. Overall the trajectories show that the larger scale, synoptic winds and coinciding wave-induced motions are the dominant driving force in the drifter movement, evident in the similarities in drifter tracts across each domain. Influence of the pre-existing regional circulation can also be observed in the variability among drifter trajectories. The undrogued trajectories seem to show less variability across the domain than for the drogued case, suggesting the wind- and wave-driven components are even more dominant in the surface layer measured by the undrogued drifters.

Scatter plots of the difference between the wind direction and the direction of the total velocity of the drogued and undrogued drifters, i.e. deflection angles, along the trajectories segments shown in black in Fig. 10, are shown in Figs. 11a and 11b, respectively. Figure 11c,d shows the deflection angles between the wind and drifter velocities, after the NCOM velocity fields of the underlying regional circulation during the high wind event (Fig. 10) have been subtracted from the full velocity of the drifters, while Fig. 11e,f show the same deflection angles, calculated using the LAVA-derived estimates of the regional circulation (Figs. 7,8). Any data points lost during the subtraction of the LAVA-derived regional circulation velocity field, shown as red segments in Fig. 10, were also excluded from the scatter plots showing the full drifter velocities (Fig. 11a, b) and the scatter plots created using the NCOM velocity fields (Fig. 11c, d) for accurate comparison. Deflection

angles are plotted against the point value wind magnitude from the UWIN-CM model, closest to the drifter location at the time of the velocity measurement.

It is evident from Fig. 11 that the resulting scatter plots, after subtracting the LAVA estimated velocities, have a more organized and compressed scatter than that of the total drifter velocity deflection angles. This is especially apparent for the drogued drifter case. The scatter plots created after the NCOM velocity fields have been subtracted from the full drifter velocity have a wider, more disorganized spread than that of the full drifter velocities, with a large portion of the deflection angles being directed to the left of the wind direction for both drogued and undrogued drifters. Resulting deflection angles using NCOM velocity fields of varying depths between 2 and 200 m, and varying in time from the timing of each LAVA velocity field observation to the end of the high wind analysis window, did not show more compressed or organized scatters. Deflection angles calculated after removing the AVISO geostrophic velocities (not shown) from the full drifter velocities, revealed little to no change compared to those of the total drifter velocities, due to the relatively small magnitude of the geostrophic velocities in the region. Because of the negligible change in deflection angle caused by the subtraction of the geostrophic velocities, and the unorganized and misaligned deflection angles resulting from the subtraction of the NCOM velocity fields, the remainder of the results only focus on those calculated using the LAVA-derived estimates for the regional circulation.

The average velocity magnitude of the calculated wind-wave-driven drifter velocities are compared to the average magnitude of the LAVA-derived pre-existing regional circulation for each wind event and drifter type in Fig. 12. The magnitude of the regional circulation is plotted as a percentage of the combined wind- and wave-driven velocity magnitude. This percentage decreases as the wind- and wave-driven effects become increasingly large during each high wind event. During the analysis periods over which we deconstruct the total surface current (shown by vertical lines in Fig. 12), the pre-existing circulation is 30-45 % as large as the combined wind-wave-driven velocity calculated using drogued drifters, and 25-30 % as large as that measured by undrogued drifters.

The wind-wave-driven, wind-driven, and wave-driven (Stokes drift) velocities are binned by wind magnitude on 0.5 m s$^{-1}$ intervals. The total number of drifter positons, per wind bin, where the drifter velocity deconstruction is performed, are plotted in Fig. 13, which shows that the most robust results exist over wind bins from 14.5-18 m s$^{-1}$ for both drogued and undrogued drifters. It should be noted that although the sample size varies considerably between drifter types and over the assigned wind bins, both drifter types have substantial sample sizes over most wind bins presented, given the relatively short time periods of analysis. The sample size of drifter measurements is also listed for each high wind event in Fig. 10. Averages and standard deviations of the velocity components were computed within each bin. The average deflection angle from the wind direction of the Stokes velocity, wave-wind-driven velocity and wind-driven velocity for each respective drifter draft depth are shown in Fig. 14. The magnitudes of the same velocity components are plotted as a percentage of the wind magnitude in Fig. 15. Figures 14c and 15c are obtained by subtracting the given Stokes drift velocity (Figs. 14a, 15a) from the estimated wind-wave-driven velocities (Figs. 14b, 15b). Bins lower than 11.5 m s$^{-1}$ have been omitted in Figs. 14 and 15,

due to lack of data points and large variation of deflection angle in the given bins. Error bars indicate one standard deviation about the mean for each bin (Figs. 14, 15).

The average wind-wave-driven component of the undrogued drifter velocity varies with increasing wind speed, traveling between ~4° and 40°, to the right of the wind, as winds increase from 11.5-20 m s$^{-1}$ (Fig. 14b). On average, the magnitude of this velocity component varies from 4.5-7.1 % of the wind speed (Fig. 15b). For the drogued drifter case, the deflection angle of the wind-wave-driven component varies from ~26°-60°, as winds increase over the same range, traveling at 2.8-4.6 % of the wind speed, on average (Figs. 14b and 15b). The total wind-wave-driven velocities exhibit larger deflection angles as wind speeds increase, with the drogued drifter derived velocity travelling at a slower speed, and being deflected ~5°-28° further to the right than that of the undrogued drifters.

Our estimates for purely wind-driven velocity components (Figs. 14c and 15c) show the undrogued drifter wind-driven velocity traveling ~5° to the right of the wind, at 6.0 % of the wind speed during 12 m s$^{-1}$ winds. The deflection angle gradually increases to ~55° to the right of the wind as wind speeds increase to 20 m s$^{-1}$. The average velocity magnitude varies between 3.4-6.0 % of the wind speed over this wind interval. The deflection angle of the drogued drifter wind-driven velocity ranges from ~30°-85°, again with higher deflection angles occurring at higher wind speeds. The velocity magnitude of this component varies from 2.3-4.1 % of the wind speed over the given wind speed interval. The difference in deflection angle between undrogued and drogued wind-driven velocities varies between ~8° and 30°, with the drogued drifter component traveling further to the right, at a slower velocity. In both cases there seems to exist an increase in deflection angle with increasing wind speed.

## 5. Discussion

Comparison of the geostrophic velocities calculated from altimetry data in Fig. 9 with SST data and the LAVA-derived velocity fields (Figs. 7,8), reveals that the regional circulation in our region of interest cannot be adequately described by geostrophic motions alone. Since the AVISO geostrophic velocities are much smaller in magnitude than the LAVA and NCOM velocity fields, it is apparent that geostrophic balance is not the dominant forcing in the region. The structures observed in the SST data and LAVA-derived velocity fields (Figs. 7,8), which have horizontal length scales of only a few tens of kilometres, are created through the interaction between the dense interior waters of the Gulf of Mexico and the less dense Mississippi River outflow waters. These features are smaller than the typical mesoscale structures resolved by the altimetry based SSH measurements, which suggests little information about the regional circulation is gained from the assimilation of the altimetry data due to the region's proximity away from the dominate mesoscale structures observed in the Gulf of Mexico. This compliments the findings of Berta et al. (2015), which found altimeter based geostrophic velocities to vary significantly from the velocity of CODE style drifters in the Northern Gulf of Mexico. This would also suggest that there are limited constraints on the modeled ocean circulation resulting from missing initial conditions away from the altimetrically-resovled mesoscale structures. Although the assimilation of altimetry data may

improve NCOM performance overall, it does not provide enough information to accurately describe the dominant dynamics of the regional circulation for the given domain. Haza et al. (2019) provides supporting results to this hypothesis, showing that during LASER the forecasted NCOM ocean velocities, agreed reasonably well to observations in the interior Gulf of Mexico where mesoscale structures dominate the flow, but were very different from observation in the Northern Gulf where smaller scale structures dominate. The increased scatter of the deflection angles, shown in Fig. 11c,d, also imply that the regional circulation is not accurately represented by the NCOM ocean velocities. The increased agreement of the structures seen in SST data and the LAVA-derived velocity fields, as well the comparison between scatter plots shown in Fig. 11, reveals that the velocity fields created using the observational drifter data are more equipped to described the regional circulation specific to this area, than altimetry based or modeled velocities.

Comparison of scatter plots in Fig. 11a,b and Fig. 11e,f indicate that the pre-existing regional circulation still influences the velocities measured by both drifter types during these large wind-driven event. The scatter of deflection angles between different drifter types also suggests that the relatively deeper layer measured by the drogued drifter exhibits larger variation in velocity due to the regional circulation component of the flow. The velocities observed with undrogued drifters seem to be more dominantly driven by the wind and wave components, as removing the influence of the regional circulation results in a less drastic compression of the scatter. In both the drogued and undrogued cases, the removal of the LAVA estimated regional circulation results in a decrease in the standard deviation of the deflection angle for all wind bins from 11.5-20 m s$^{-1}$.

Comparison of the relative magnitudes of the combined wind-wave-driven velocities and the pre-existing circulation estimates confirms that the drifter velocities are dominantly wind- and wave-driven during the high wind periods (Fig. 12). Analysis of the plots in Fig. 12, led the authors a-posteriori, to determine the optimal windows for the velocity deconstruction results during the high wind events. With respect to the subtraction of the pre-existing regional circulation and the improvement of the scatter in deflection angles (seen in Fig. 11e, f), the most robust results, for both drifter types, occur when the relative strength of the wind-wave-driven flow to the pre-existing regional circulation reaches a plateau (seen in Fig. 12). This is also when the wind, and therefore the wind-wave-driven component, is the strongest, which suggests that any error introduced during the velocity deconstruction makes up a smaller percentage of the calculated velocities. Extending the velocity deconstruction analysis windows, by 2 hours before and 2 hours after, does not significantly change the averages and trends shown in Figs. 14 and 15, but does however, slightly disorganize the scatter of the deflection angles seen in Fig. 11e, f. This investigation of sensitivity also aided in determining the optimal time periods for analysis.

Figure 14b shows that the estimated deflection angle of the combined wind- and wave-driven velocities of drogued and undrogued drifters, vary significantly with increasing wind velocity. Since the deflection angle of the Stokes drift velocity (Fig. 14a) is almost constant with wind speed, the increase in deflection angles seen in Figs. 14b, is most likely due to a change in the wind-driven momentum input. The increased deflection angle observed between the two drifter types seems to be predominately wind-driven based on the relative magnitudes of Stokes drift and estimated wind-driven currents (Fig 15). Classical Ekman theory, is based on the balance between Coriolis and vertical viscosity in the water column, which

given the parameterization for wind stress and viscosity assigned by Ekman (1905), results in a wind-driven surface current deflected 45° to the right of the wind in the Northern Hemisphere, which spirals to the right and decreases in magnitude with depth. In contrast to Ekman, the "Slab" solution, based on enhanced surface mixing due to breaking waves and shear-induced turbulence, prescribes a linear decrease of wind stress with depth, resulting in a surface current which travels 90° to the right of the wind, uniformly with depth (Pollard and Millard, 1970).

The findings presented here, seem to support aspects of both the Ekman and Slab solutions, as there does appear to be a rotation of the wind-driven drifter velocities with depth, but overall, these velocities display a larger deflection than predicted by Ekman at larger wind speeds. This can possibly be explained by enhanced vertical mixing under high wind and wave conditions, which acts to diminish the effects of stratification and distribute wind-driven momentum into the water column. This theory motivates the assignment of a linearly decreasing wind stress with depth used in the slab model solution by Pollard (1970) which results in deflection angles of 90 degrees over all depths of the mixed layer. The observed increase in deflection angle with wind speed may suggest a gradual change in wind-driven flow regimes, from surface Ekman dynamics to more Slab-like dynamics, as increasing wind velocity and subsequent turbulence from vertical shear and breaking waves, mix momentum vertically in the wind-driven layer. However, there still exists a rotation of the wind-driven current with depth which cannot be explained using slab-like dynamics alone.

Comparison to more recent observational studies is nontrivial due to differences in instrumentation, measurement depth, and time resolution of measurements. Many previous studies have performed spectral analysis on years of HF radar or drifter data with time resolutions varying from 0.5-6 hours, and calculated correlations between wind and surface currents in order to isolate the wind-driven component of the flow. Results from these studies find a range of deflection angles from 17-90 degrees to the right of the wind, with current magnitudes varying from 0.4-5 % of the wind speed (Ardhuin et al., 2009; Poulain et al., 2009; Kim et al., 2010; Röhrs and Christensen 2015;). Sentchev et al. (2017) used instantaneous vertical ADCP measurements to calculate the wind-driven effect on surface currents (0.5-75m depth) during sea breeze conditions (wind speeds of ~3-6 m s$^{-1}$), finding wind-driven deflection angles of 12°-25° and current magnitudes of 2-4 % of wind speed. The study also found a rotation to the right was present between wind-driven velocities of increasing measurement depth. Berta et al. (2018) used hourly HF radar measurements to observe the influence of large wind events lasting 1-3 days. After subtracting the geostrophic component of the flow to isolate the wind-driven current, they found surface velocity deflection angles and magnitudes of 25°-30° and ~2 % of the wind speed, respectively. More details of these previous studies on wind-driven dynamics can be found in Table 1.

The findings portrayed in this paper seem to be within the range of previously reported results for wind-driven surface flows with discrepancies likely resulting from the differences in the depth of the measurement, range of wind magnitudes, upper ocean stratification, or methodology in isolating the wind-driven component. Several of these previous studies also utilized data sets with a portion of surface current measurements occurring close to the coast, were topography could've added noise to the measurements (Poulain et al., 2009; Kim et al., 2010; Röhrs and Christensen, 2015; Berta et al., 2018). With the exception of Ardhuin et al. (2009), these past studies have not accounted for Stokes drift in their

measurements, sometimes due to negligible wave influence under low wind conditions, possibly resulting in smaller deflection angles due to the near alignment of winds and wave direction (Poulain et al., 2009; Kim et al., 2010; Röhrs and Christensen 2015; Sentchev et al., 2017; Berta et al., 2018). The UWIN-CM modeled Stokes velocity used for this study, seems to be in good agreement with the full Stokes velocity magnitude and deflection angle, with respect to the wind, presented by the previous literature (Ardhuin et al., 2009).

We explore the possible influence of wind-driven effects during the observation of the pre-existing regional circulation, by conducting a sensitivity test, where an estimated wind-driven velocity (based on the UWIN-CM wind velocities) is removed from the LAVA-derived velocity fields before the total velocity deconstruction is performed. Relying on previous literature, we estimate that during low wind conditions, the undrogued drifters travel at 3.5 % of the wind, being deflected 5° to the right, while the drogued drifters travel at 1.5 % of the wind, being deflected 25° to the right. These values were chosen, based on the results of all the literature shown in Table 1, to represent moderate wind-driven effects on the surface drifters. Laboratory testing and field experiment results using the CARTHE drifters, also aided in assigning the differential velocities between drifter type (Laxague et al., 2017; Novelli et al., 2017). Performing the total velocity deconstruction with this estimated wind-driven effect removed from the pre-existing circulation, on average, results in a decrease of wind-driven velocities magnitudes by 0.58 % and 0.34 % of the wind velocity for undrogued and drogued drifters, respectively. The average deflections angles of the wind-driven component of both drifter types, become more aligned with the wind by about 5° for drogued drifters and 7° for undrogued. While there is some change in the final magnitudes and deflection angles calculated, the overall trends seen in Figs. 14 and 15 do not change as a result of this sensitivity analysis. Additionally, the scatter plots of deflection angles (Fig 11e,f) become slightly more scattered (standard deviation slightly increased) with this estimated wind influence removed, and become even less organized when the estimated wind-driven velocities during the low wind periods are increased further. Because the true values of the wind-driven velocity components during these low wind-periods are not well known, and the standard deviations of deflection angle are slightly increased by accounting for such, we do not account for any wind-driven influence during the low wind periods for the final results. The assignment of decreasing velocities away from the drifter locations up to a length R, prescribed by the LAVA algorithm, in addition to averaging, acts to smooth out the absolute velocities of the individual drifters, which adds complexity to accounting for wind-driven effects during these low wind periods.

Another source of error in the wind-driven measurements reported here is the possible evolution of the regional circulation during the time gap between the observation of the pre-existing regional circulation and the velocity deconstruction during the high wind period. Although difficult to quantify given the available data, the regional circulation is evolving, to a certain extent, during the time gap between the observation of the regional circulation and the analysis periods of high winds. In addition, the regional circulation could begin to be modified through interactions with the large wind- and wave-induced currents. The extent to which this is occurring can be seen qualitatively in Figs. 7 and 8 by comparing the change of the SST fields in time. One example of which is evident in the South-eastward progression of the front seen in the SST fields associated with the January high wind event. The movement of this front seems to generally mimic the movement

of the drifters during this high wind event shown in the top panels of Fig 10. The overall timescales on which these flow features evolve or are altered by each high wind event is beyond the scope of this paper, given the available data set. However, the relatively short time scales of these periods of high winds and the overall agreement between the pre-existing circulation estimates and the SST fields (shown in Figs. 7 and 8) suggest that the method of velocity deconstruction used here is adequate. In addition, the decreased variability seen in the deflection angles after the subtraction of the pre-existing regional circulation (Fig. 11) suggests, a-posteriori, that the regional circulation does maintain its structure to a reasonable extent during the periods of increasing winds being analyzed.

Another possible source of error comes from the velocity slip characteristics of the drifters mentioned above, mostly relevant in the undrogued case. The magnitude of the wind-wave-driven and purely wind-driven velocities of the undrogued drifters are among the higher estimates previously reported (Ardhuin et al., 2009; Poulain et al., 2009; Kim et al., 2010; Sentchev et al., 2017; Berta et al., 2018). The effect on the velocity slip of these drifters due to windage and wave steepening has been tested in the lab by Novelli et al. (2017). Although there exists a large difference in scales between the lab and the real ocean, the differences between the combined wind-wave-driven velocity magnitudes of drogued and undrogued drifters (~2 % of the wind speed on average) in the current study, are in good agreement to that of laboratory studies using half scale drifters and a past field experiment using full size drifters (Laxague et al., 2017; Novelli et al., 2017). As mentioned above, laboratory testing of undrogued drifters showed that the total velocity slip could be as high as 14 cm s$^{-1}$, but was shown to decrease to only 1 cm s$^{-1}$ with increasing wind speeds (over a range of 15-23 m s$^{-1}$ in the lab) due to the sheltering of drifters from the wind by increasing wave heights (Novelli et al., 2017). This could partly explain the gradual decrease in wind-driven velocities for both drifter types from wind bins 11.5-18 m s$^{-1}$, especially given the relatively high wind-driven velocities seen at lower wind bins. Exactly how much wind-slip is occurring during specific wind velocities and wave heights in the real ocean is difficult to determine, but the average magnitude of the observed vertical shear seems to be in relatively good agreement with past experiments and laboratory testing performed with these drifters (Laxague et al., 2017; Novelli et al., 2017).

The momentum input from large breaking waves into the surface currents at the very high wind speeds studied here, could also cause an increase in velocities observed in the wind-driven surface currents estimated from either drifter type. In addition, the surfing behavior of the undrogued drifters could amplify this increase in observed velocities further. Velocity slip due to wind and breaking waves could also account for some of difference seen in deflection angles between the drifter types, as the influence of wind and waves would keep the undrogued drifters more in-line with the wind. Ardhuin et al. (2009) attributes larger deflection angles of wind-driven currents to enhanced mixing due to wave breaking, which could be a congruous theory to our observation of increasing deflection angles at higher wind speeds. Enhanced mixing caused by breaking waves, acts to mix the vertical momentum of surface currents, likely resulting in larger deflection angles at shallower depths (Rascle et al., 2006).

The results presented here, to the authors' knowledge, are among the few studies able to reported estimates of the vertical shear of wind- and wave-driven velocity components in the upper 1 meter of the ocean under this regime of high winds (11.5-20 m s$^{-1}$). The combined wind-wave-driven velocity of the undrogued drifters calculated here is, on average, ~1.6 times greater than that measured by drogued drifters for the wind magnitudes presented. These results support the finding of Laxague et al. (2017), which showed that, in the presence of negligible stratification in the upper layer, total surface currents in the upper 1 cm are twice as fast as the average current over the first 1 m (measured to be 0.57 m s$^{-1}$ and 0.3 m s$^{-1}$, respectively) due to wind- and wave-driven vertical velocity shear. It should also be noted, however, that vertical shear in the presence of strong mixing (i.e. wave breaking) acts to diminish vertical velocity gradients, which could suggest some of the shear observed is due to the velocity slip of the undrogued drifters (Sutherland et al. 2016). The true novelty of the results presented here lies in the instantaneous estimates for the vertical shear of the purely wind-driven velocities calculated each drifter type, made possible through the method of velocity deconstruction used here. The deconstruction of the total velocity measured from drogued and undrogued drifters gives us an estimate, directly related to the wind velocity, of the wind-driven vertical shear at the very surface of the ocean, which has proved difficult to measure by previous studies and has significant implications for surface transport problems in the real ocean.

The high wind events focused on here only occurred for a small number of days during the LASER campaign, which is typical for the Northern Gulf of Mexico in the winter. Since the velocity profile within upper meter has been shown to be very dynamic, being affected by the general oceanic circulation as well as local wind-wave-driven mechanisms, it is important to observe near surface currents under different environmental conditions and at greater vertical resolution. Attention also needs to be given to the transport of Lagrangian particles by breaking waves which induce vertical motion and mixing which can alter wind-driven currents.

**6. Conclusion**

We use a combination of stored output data from the UWIN-CM fully coupled atmospheric-wave-ocean model and observational trajectory data from both drogued and undrogued CARTHE drifters to calculate an estimate for purely wind-driven drifter velocities during periods of strong, increasing winds. The use of co-located drogued and undrogued drifters provides measurements for the vertical shear between the upper 5 cm and upper 60 cm surface layers. Using the LAgrangian Variation Analysis, we are able to create velocity fields in the hours leading up to the high wind events studied, that serve as an estimate for the pre-existing regional circulation which is found to still affect the drifter velocities during periods of high winds. After subtracting the regional circulation from our measured drifter velocities we analyze the relationship between wind velocity, Stokes drift, and wind-driven velocity of each drifter type. On average, we find the wind-driven velocity component between drifter types to decrease in magnitude and rotate to the right of the wind with depth, with the undrogued (drogued) component traveling ~5°-55° (~30°-85°) at 3.4-6.0 % (2.3-4.1 %) of the wind speed over the range of 12-20 m s$^{-1}$.

Both wind-driven velocities display an increase in deflection angle with increasing wind speed, sustaining an average difference of 8°-30° between deflection angles.

This study is among the few (Sentchev et al., 2017; Berta et al., 2018) that have investigated the real-time response of very near surface currents to increasing wind. We are able to observe characteristics of the vertical shear present between the upper 60 cm (drogued) and 5 cm (undrogued) of the wind-driven component of the drifter velocities. Observations of vertical shear this close to the surface in the real ocean, especially during high wind events of this nature, are very scarce due to the limitations of present day instrumentation. The vertical velocity profile within the upper meter has been shown to exhibit a large amount of shear, with velocities at the surface (upper few centimeters) being largely dominated by wind and waves (Laxague et al., 2017; Morey et al., 2018). This highlights the importance of very near surface observations, such as those presented in this paper, as this vertical shear can have large impacts on the transport of pollutants of varying size and buoyancy, like plastics and oil. Incorporating vertical shear due to wind and waves within the upper meter, could have important implications for fate prediction of pollutants transported at the ocean surface.

**Data Availability.** Raw, processed drifter trajectory data, and drogue classification are publicly available through the Gulf of Mexico Research Initiative Information & Data Cooperative (GRIIDC) at https://data.gulfresearchinitiative.org under https://doi.org/10.7266/N7MS3R6V, https://doi.org/10.7266/N7W0940J, and https://doi.org/10.7266/N7QN656H, respectively. The UWIN-CM data can also be obtained from GRIIDC under https://doi.org/10.7266/N7KW5DH7, where the moving vessel profiler CTD data can also be found under https://doi.org/10.7266/N7H130FC. AVISO SST and absolute geostrophic velocity products, produced by Collecte Localisation Satellites (CLS) with support from Cnes, are available upon request.

**Competing interests**. The authors declare that they have no conflict of interest.

**Author Contribution.** The formal analysis was carried out by JL as a PhD candidate under the supervision of TO. All authors contributed in the conceptualization of the work. MB educated JL on the necessary algorithm (LAVA) used to performed the formal analysis.

**Acknowledgements**. This research was made possible by a grant from The Gulf of Mexico Research Initiative. The authors would like to thank all those involved in planning and executing LASER, as well as the scientists involved with the data processing and drogued detection that made this analysis possible. The authors would like to express their gratitude to Dr. Milan Curcic for sharing his expertise of the UWIN-CM.

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

| Literature | Instrument Used | Measurement Depth | Wind Magnitude | Avg. Velocity of Wind-Driven Current (% of Wind) | Avg. Deflection Angle to the Right of Wind |
|---|---|---|---|---|---|
| **Ardhuin et al. 2009** | HF radar | 0-1 m | 1-19 m s$^{-1}$ | 0.4 – 0.8 % | 45° - 70° |
| **Kim et al. 2010** | HF radar | 0-1 m | 1-5 m s$^{-1}$ | 2 – 5 % | 50° - 90° |
| **Berta et al. 2018** | HF radar | 0-0.75 m | 10-20 m s$^{-1}$ | 2 % | 25° - 30° |
| **Sentchev et al. 2017** | ADCP | 0.5-0.75 m | 1-6 m s$^{-1}$ | 2 – 4 % | 12° - 25° |
| **Poulain et al. 2009** | Undrogued SVP drifter CODE drifter | 0-0.25 m 0-1 m | 0-15 m s$^{-1}$ | 2 % 1 % | 17° - 20° 28° - 30° |
| **Röhrs and Christensen, 2015** | iSphere CODE drifter | 0-0.15m 0.3-1 m | 0-20 m s$^{-1}$ | 3-5% 2.4% | 64° 84° |
| **Current Work** | CARTHE Undrogued CARTHE Drogued | 0-0.05 m 0-0.60 m | 11-20 m s$^{-1}$ | 3.6-6 % 2.3-4.1 % | 5° - 55° 30° - 85° |

**Table 1: Review of literature on wind-driven surface currents using real ocean, observational data.**

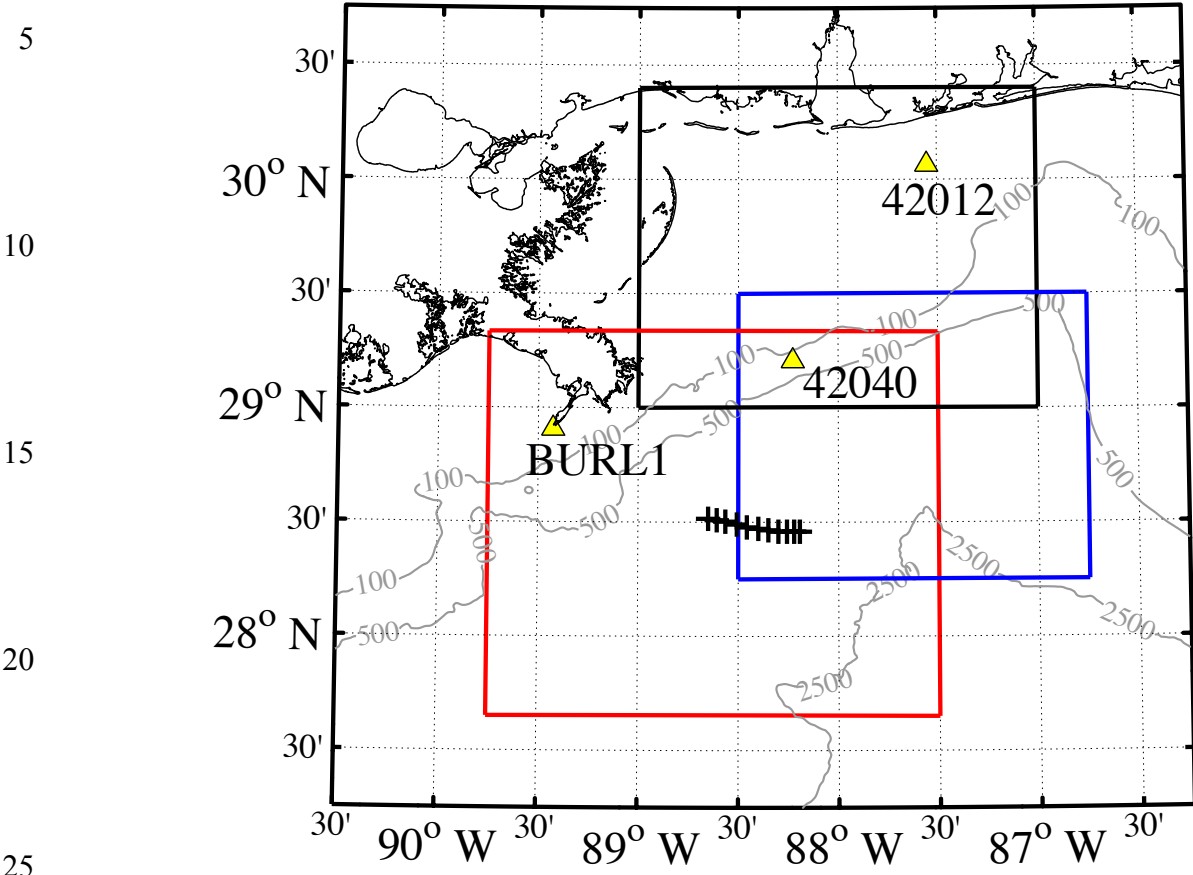

**Figure 1: Research area in the Northern Gulf of Mexico, ESE of the Mississippi River delta. NDBC buoy locations used for validation of UWIN-CM wind and wave data are shown by yellow triangles. Domains corresponding to each high wind event analyzed are shown in the red (February 24[th]), blue (January 22[nd]), and black (March 20[th]) boxes. Black hash marks show the ship track where data in Fig. 2 were collected. Grey contours show the bathymetry at 100, 500 and 2500 meters.**

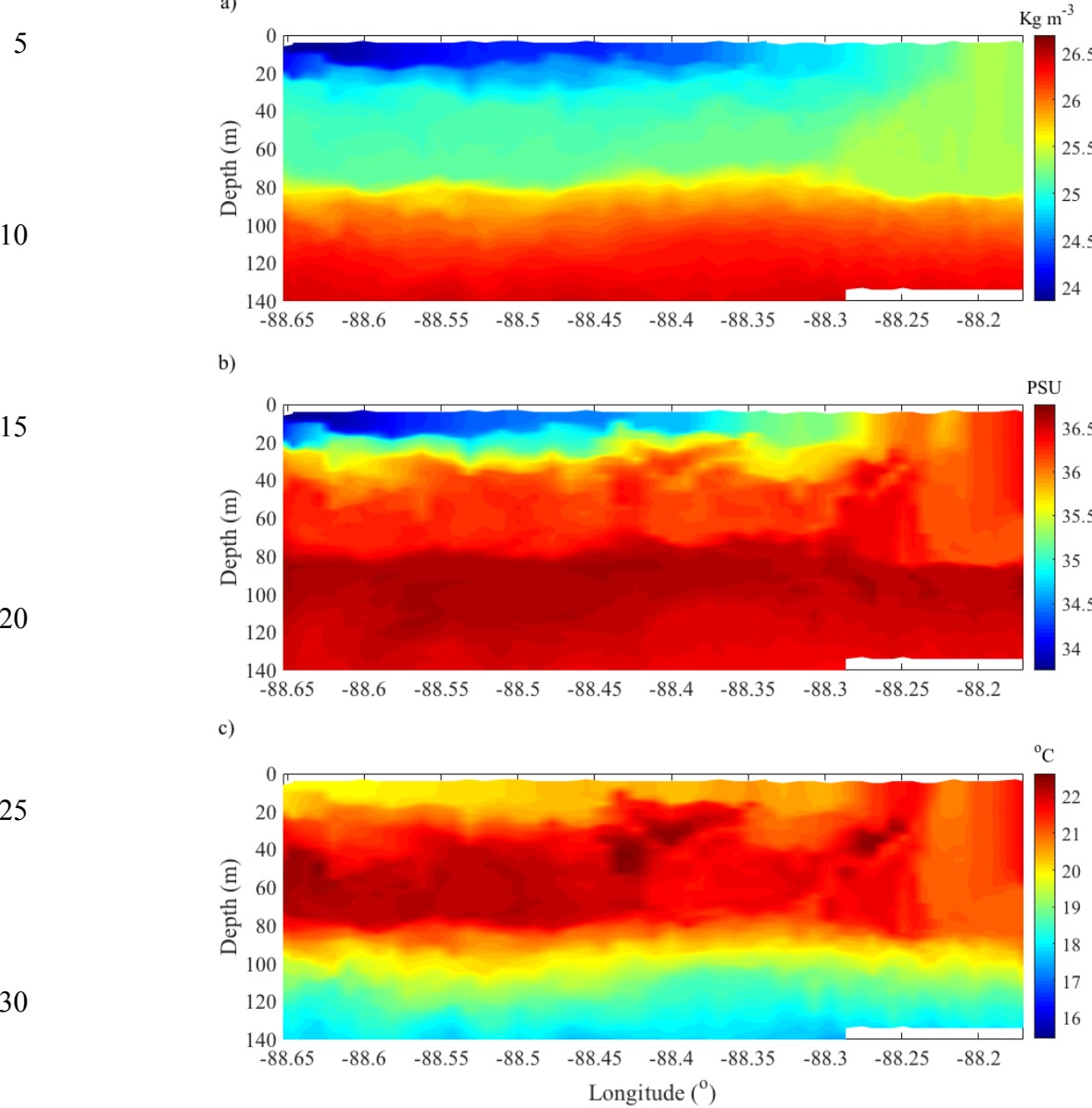

**Figure 2: Potential density (a), salinity (b) and temperature (c) data along the eastward traveling transect shown in Fig. 1, measured by moving vessel profiler (MVP) CTD casts taken on January 28<sup>th</sup> from 16:48 – 21:00 UTC.**

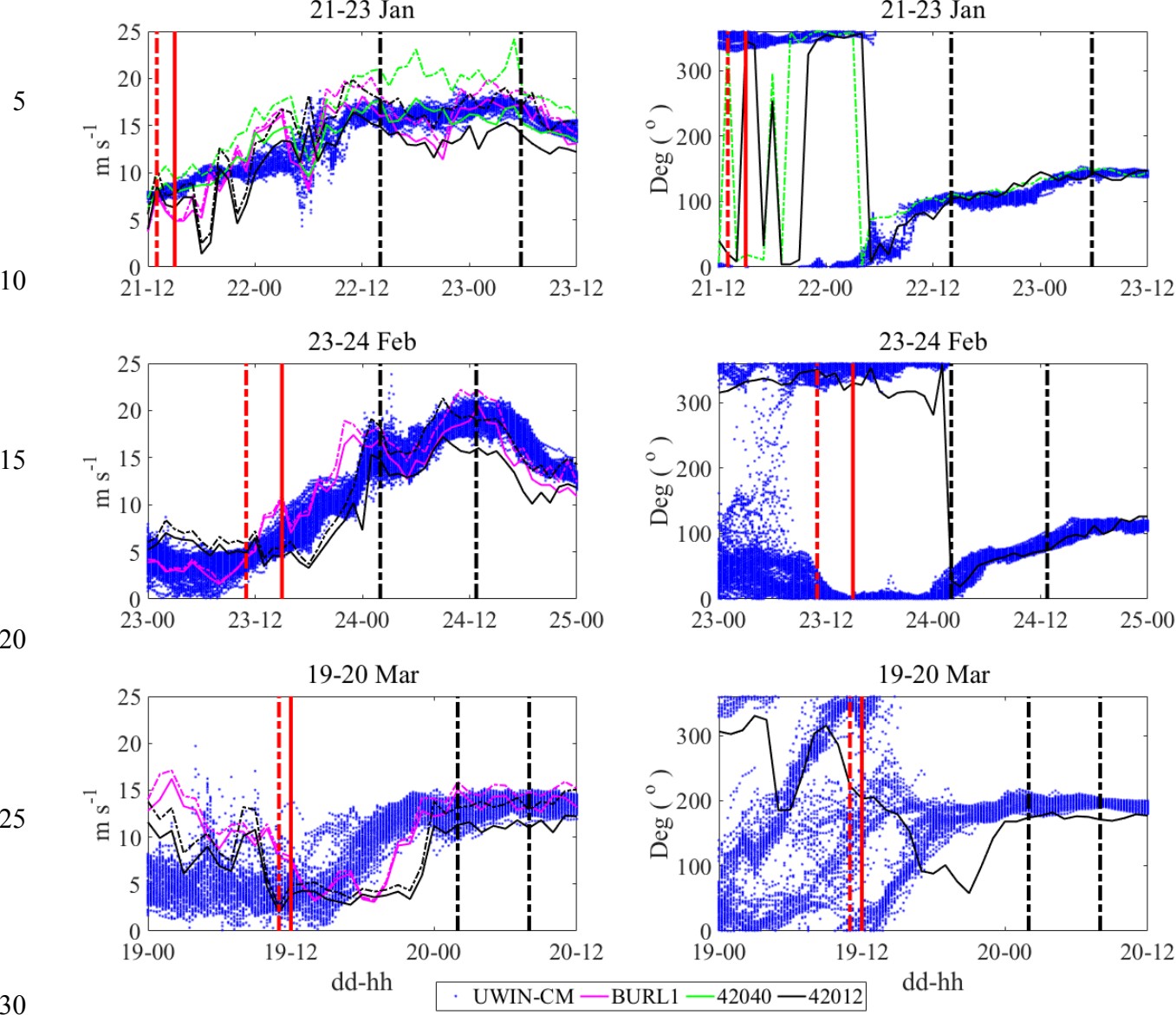

**Figure 3: Wind velocity magnitudes (left) and directions (right) during the hours preceding, and including, each high wind event. Black dashed vertical lines show the beginning and end of the high wind analysis periods. Red dashed(solid) red lines indicate beginning of the hour over which the pre-existing regional circulation was estimated with undrogued(drogued) drifters. U10 wind data from the UWIN-CM are plotted at the nearest point to every drifter location. Wind observations from NDBC buoys BURL1, 42040, an 42012 are plotted as well. Solid lines show sustained winds, while dashed lines show wind gusts. All wind directions are plotted as the direction in which the wind is traveling. Buoy 42040 was only operational during the first wind event and BURL1 did not record any wind direction data during the experiment.**

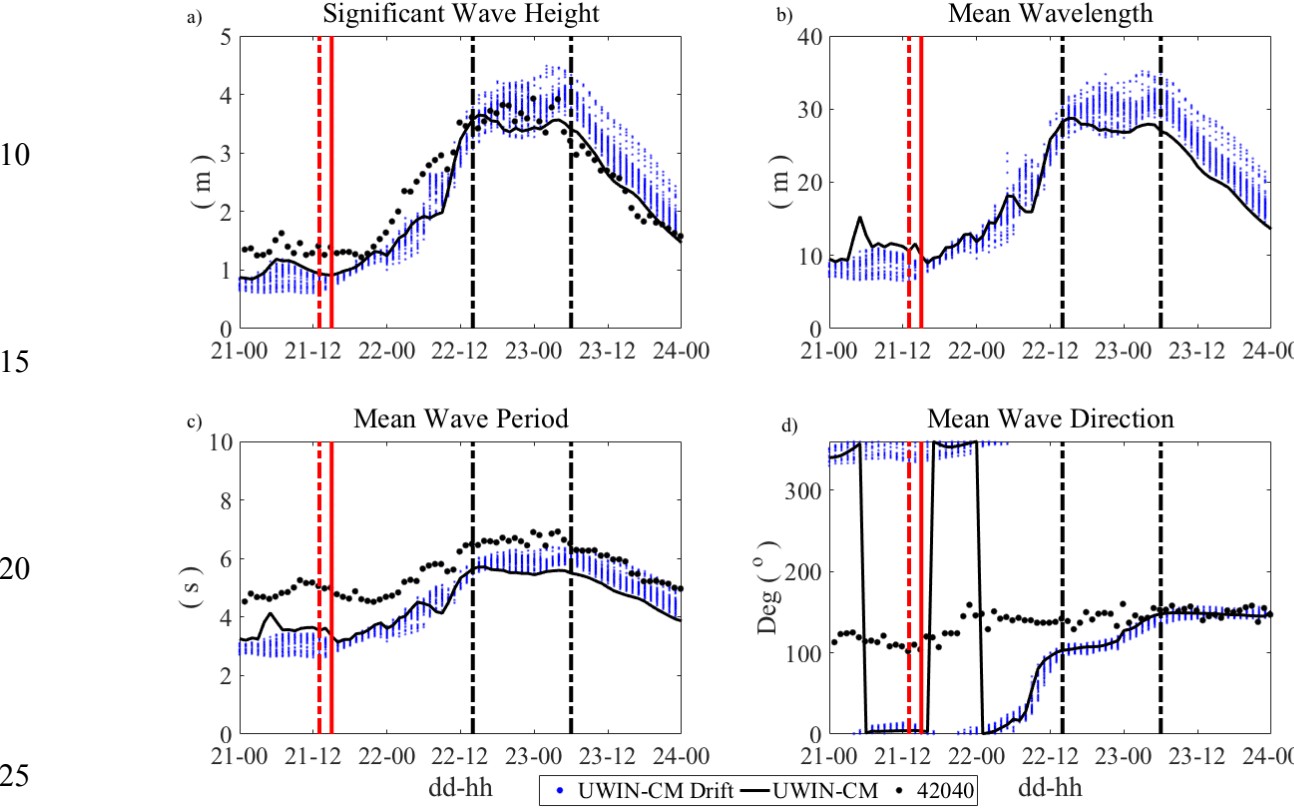

**Figure 4: Significant wave height (a), mean wavelength (b), mean wave period (c) and mean wave direction (d) from the UWIN-CM and NDBC buoy 42040 during the January high wind event. Black dashed vertical lines show the beginning and end of the high wind analysis periods. Red dashed(solid) red lines indicate beginning of the hour over which the pre-existing regional circulation was estimated with undrogued(drogued) drifters. Wave data from the UWIN-CM, plotted at the nearest point to every drifter location, are shown in blue. The solid black lines show the UWIN-CM output data closest to the coordinates of buoy 42040, while observations from buoy 42040 are shown black dots. Wavelength data was not collected by this NDBC station.**

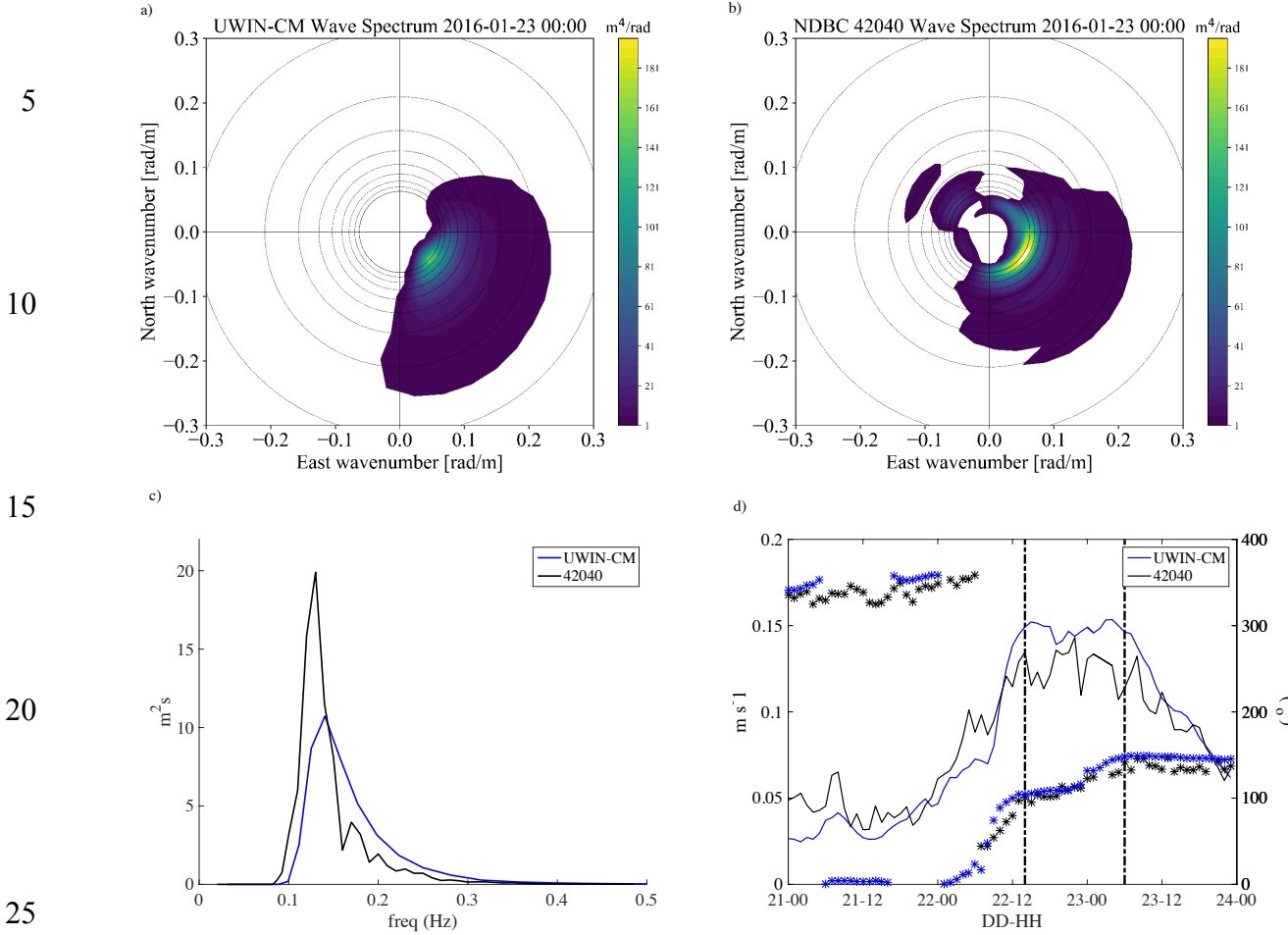

**Figure 5: 2-D wave energy spectrum from (a) UWIN-CM and (b) NDBC buoy 42040 on January 23[rd] at 00:00 UTC. Circles around the center of the plots show wavelengths ranging from 100 m (smallest) to 20 m (largest) on 10 m intervals. (c) 1-D wave density spectrum from UWIN-CM and NDBC from the same time. (d) Time series of Stokes drift velocity magnitude (solid lines) and direction (asterisk) calculated from the 2-D wave spectrum from buoy 42040 and the UWIN-CM model using Eq. 2. Vertical black lines indicate period of velocity deconstruction. The UWIN-CM Stokes drift velocity plotted here is integrated from the surface down to 1.68m, which is roughly the depth of the SCOOP style NDBC buoy used at station 42040. All data from UWIN-CM is extracted from the nearest point to the location of buoy 42040.**

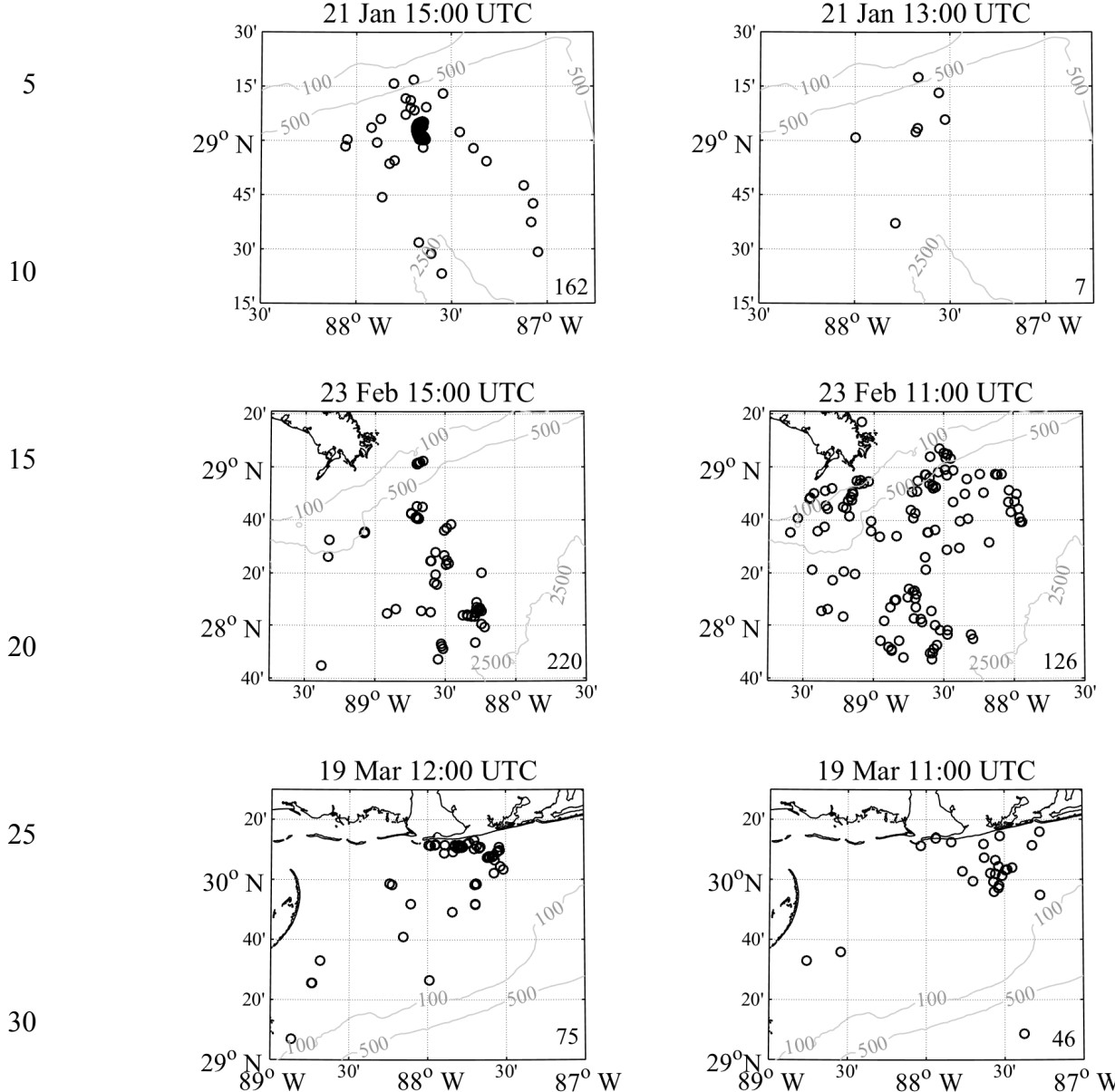

**Figure 6: Drogued (left) and undrogued (right) drifters used to create Eulerian velocity fields of the pre-wind event regional circulation estimates using the LAgrangian Variational Analysis (LAVA). Raw drifter locations are plotted at the end of the hour over which the velocity fields are created. Numbers in bottom right corner of plots display the number of drifters used for each velocity field construction. Panels correspond to sub-domains shown in Figure 1.**

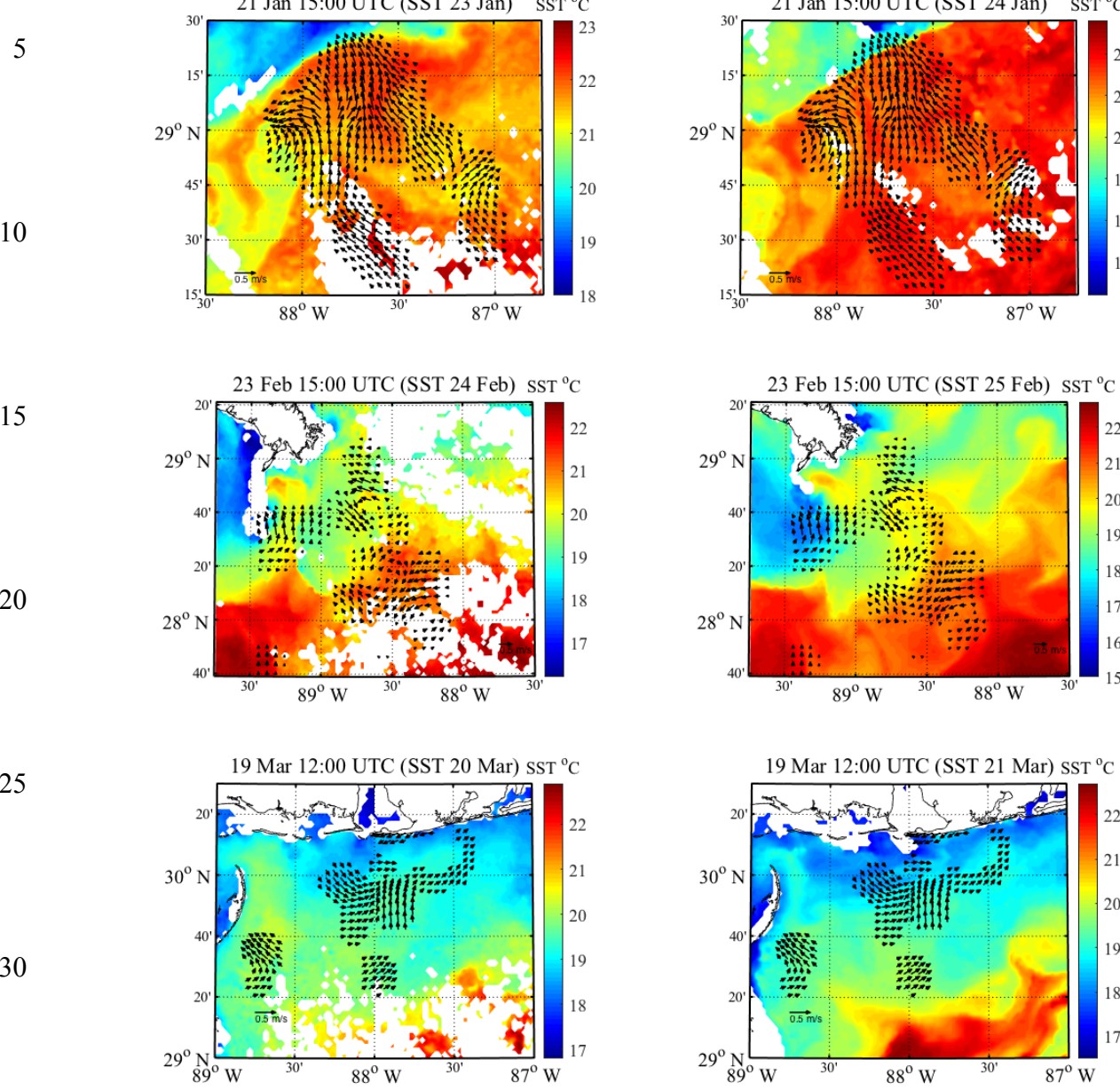

**Figure 7: Eulerian velocity fields of the regional circulation that preceded each high wind event, created using LAVA for drogued drifters in the area. 24-hour averages of AVISO SST data during (left) and after (right) each high wind event are also plotted. Titles of each plot list the beginning of each hour over which the velocity field was created from drifter trajectories and the day corresponding to the AVISO SST data. 24-hour averages, produced by AVISO, are centered on the $0^{th}$ hour of the day listed. In each velocity field, every third vector is plotted for visibility. Velocity fields on the left and right side panels are identical.**

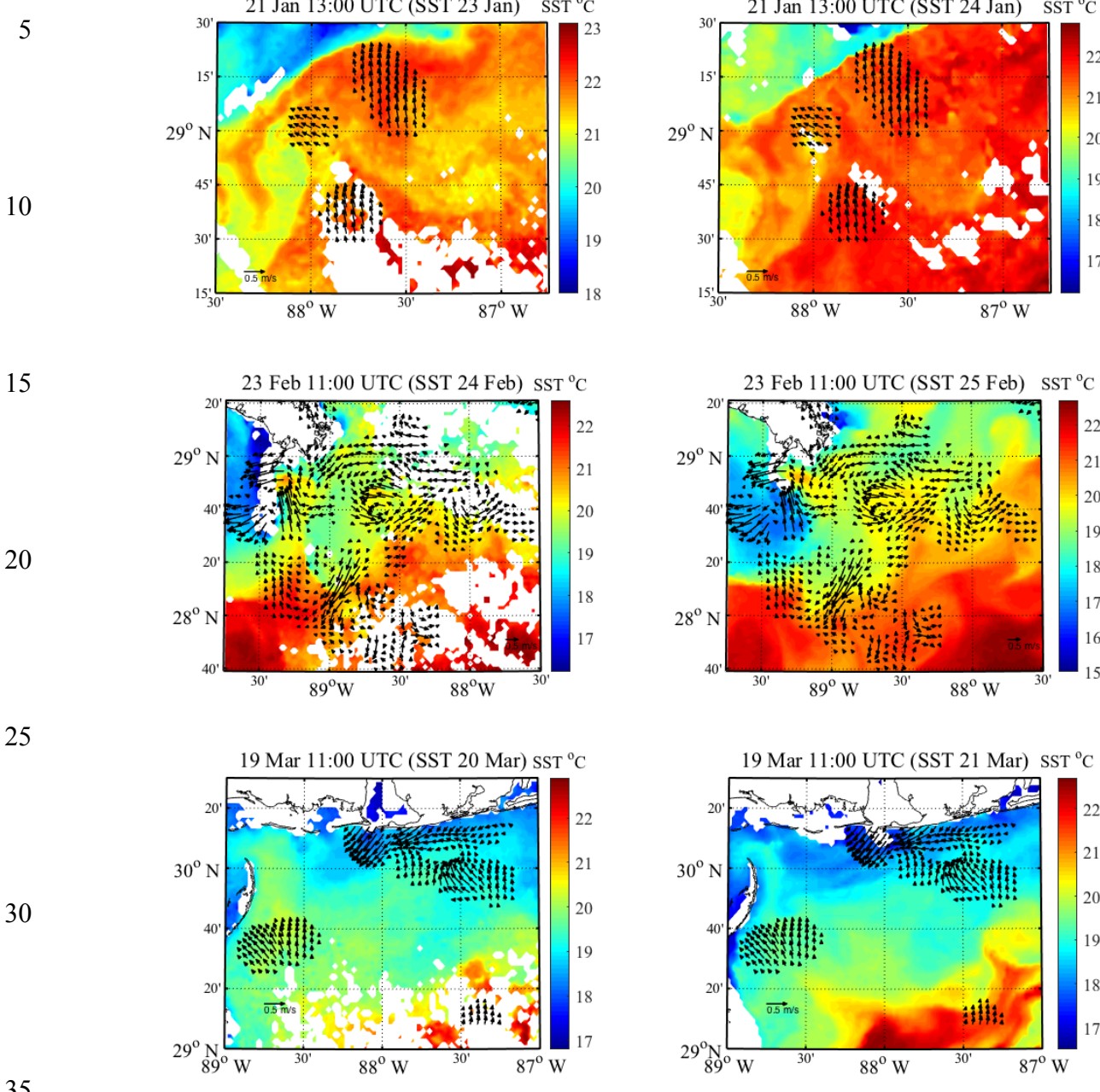

**Figure 8: Eulerian velocity fields of the regional circulation that preceded each high wind event, created using LAVA for undrogued drifters in the area, plotted on top of the same 24-hour averages of SST data shown in Fig. 7. Titles and data are plotted in the same manner as Fig. 7.**

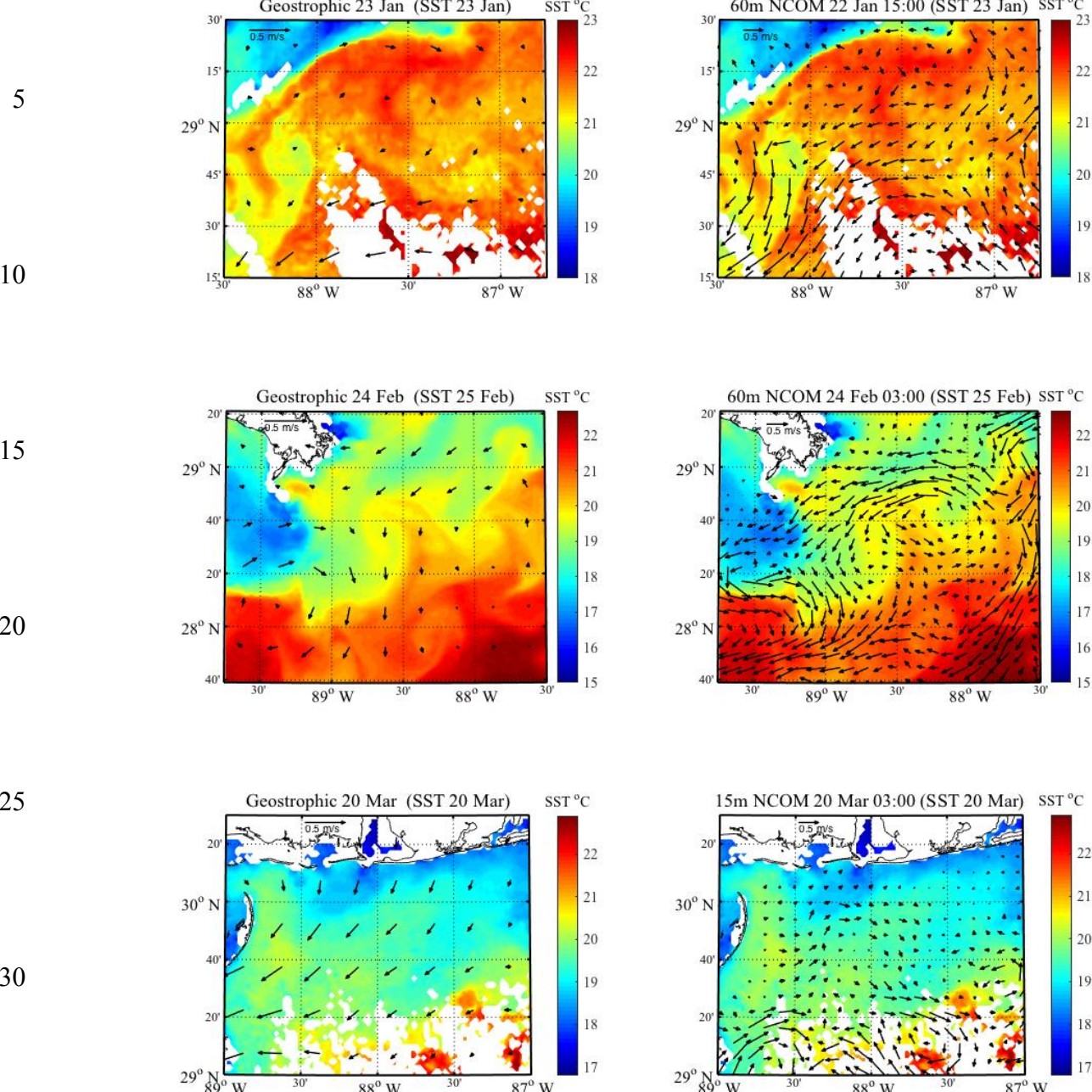

**Figure 9: AVISO based geostrophic velocities (left) and NCOM velocities (right) associated with each high wind event, along with relevant SST data. SST maps were chosen based on coinciding timing with each high wind event and/or visibility of circulation patterns. The date of each daily geostrophic velocity or 3-hour NCOM velocity field are listed for each panel. Dates of SST data are listed in the same manner as Figs. 7 and 8. NCOM velocities are plotted at 60m depth for the January and February events, while at 15m for the March event. Velocity arrows for geostrophic velocities are scaled twice as large than those of NCOM fields.**

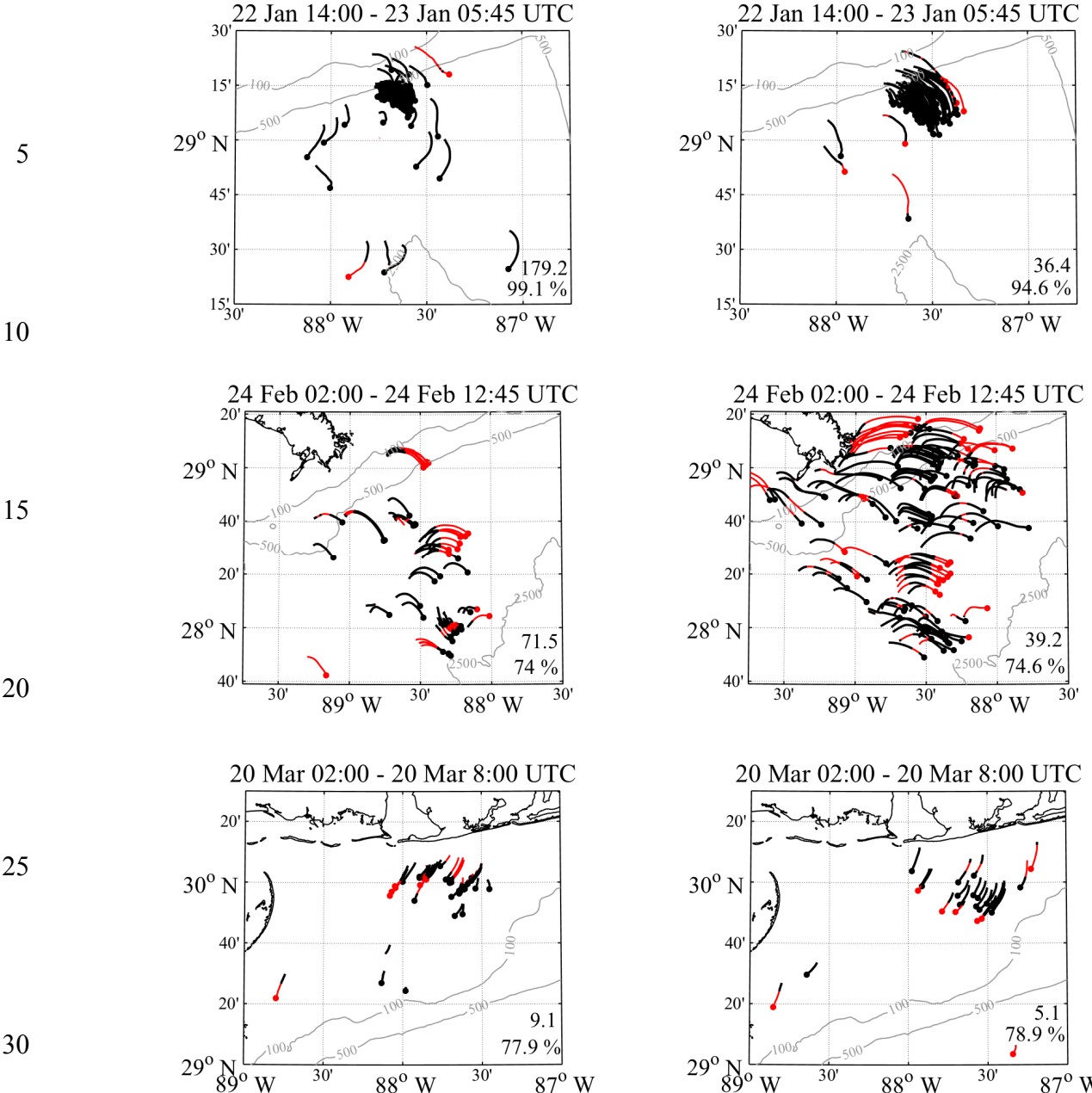

**Figure 10: Drogued (left) and undrogued (right) drifter trajectories during the hours of analysis for each high wind event. Time periods over which the total velocity deconstructions were calculated are shown in each title. Portions of the trajectories shown in black (red) depict segments which overlap (do not overlap) spatially with each respective LAVA-derived Eulerian velocity field in Figs 7 and 8. (Top) the total number of retained trajectory data points, defined as number of drifter days given 15-minute time steps, and (bottom) the percentage of coverage between trajectories and each respective Eulerian velocity field in Figs. 7 and 8 are listed within each plot. Trajectories missing head markers are a result of either drogue loss or lost GPS transmission during the experiment.**

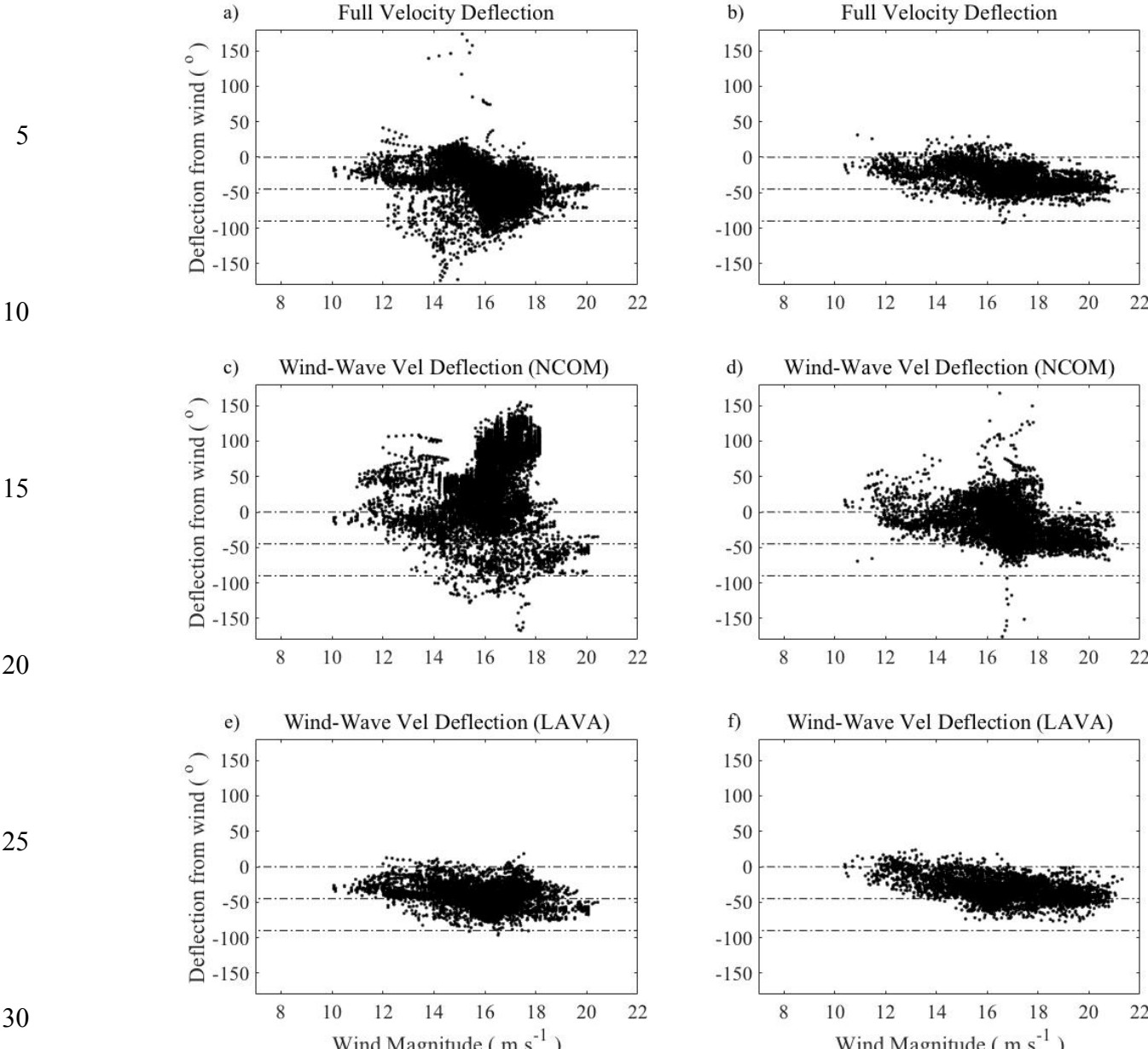

**Figure 11: Scatter plots of deflection angles of drogued (a,c,e) and undrogued (b,d,f) drifter velocities before (a,b) and after (c,d,e,f) subtracting estimates for the regional circulation during the high wind periods analyzed. Panels c and d are created using the NCOM velocity fields during the high wind events (Fig. 9), while panels e and f use the LAVA-derived estimates observed before each high wind event (Figs. 7 and 8). Data points that do not have both individual drifter velocity measurements and coinciding data in the LAVA-derived Eulerian velocity fields of the pre-existing circulation during the high wind event are not included in any panel. Deflection angles are plotted against the U10 wind magnitude from the UWIN-CM model at the nearest point to the drifter locations in the domain.**

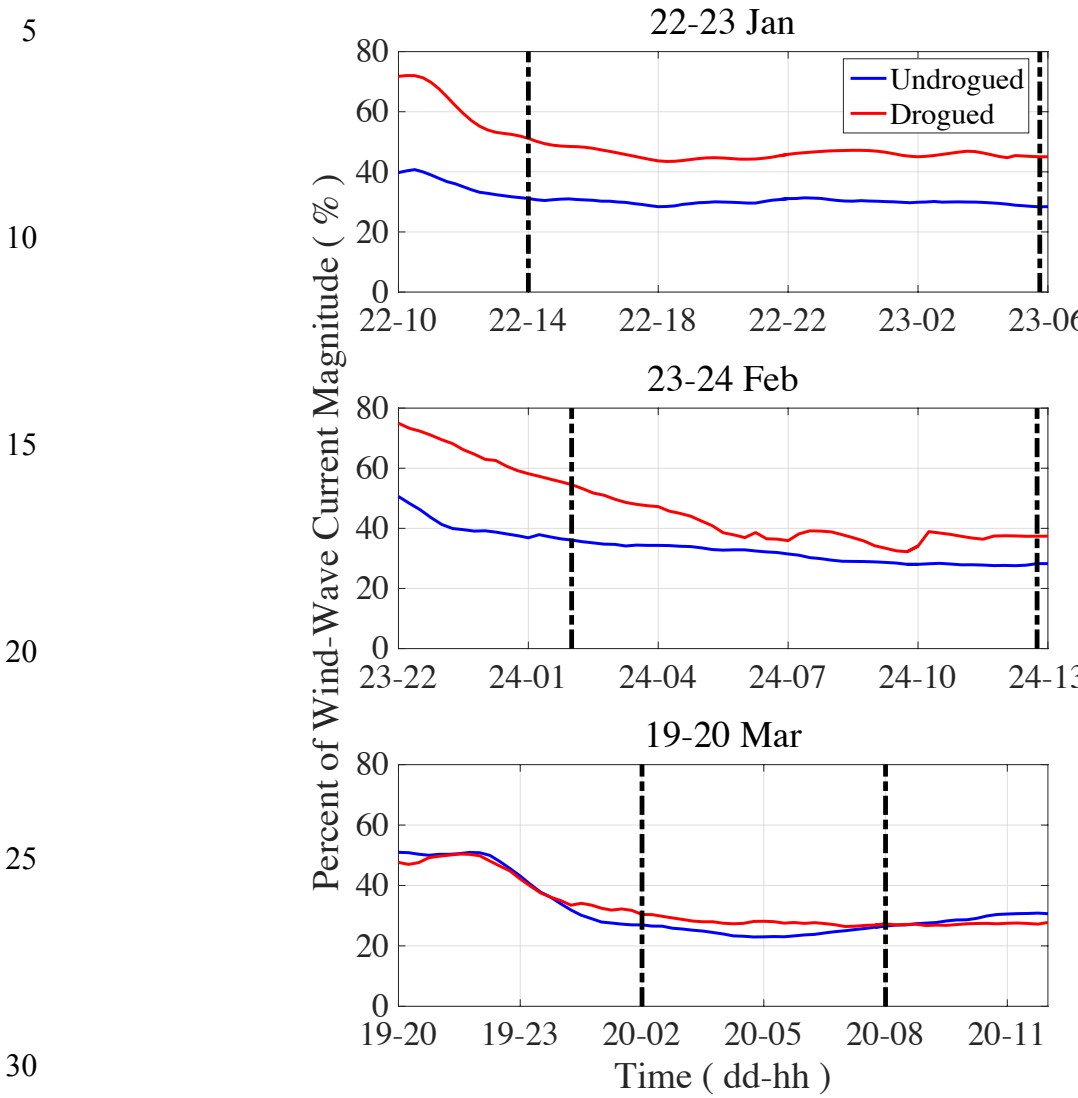

**Figure 12: Average of the Eulerian LAVA-derived velocity field estimates of the pre-wind regional circulation plotted as a percentage in time of the average combined wind- and wave-driven flow for drogued and undrogued drifters. Vertical dashed lines depict the analysis period for the total velocity deconstruction.**

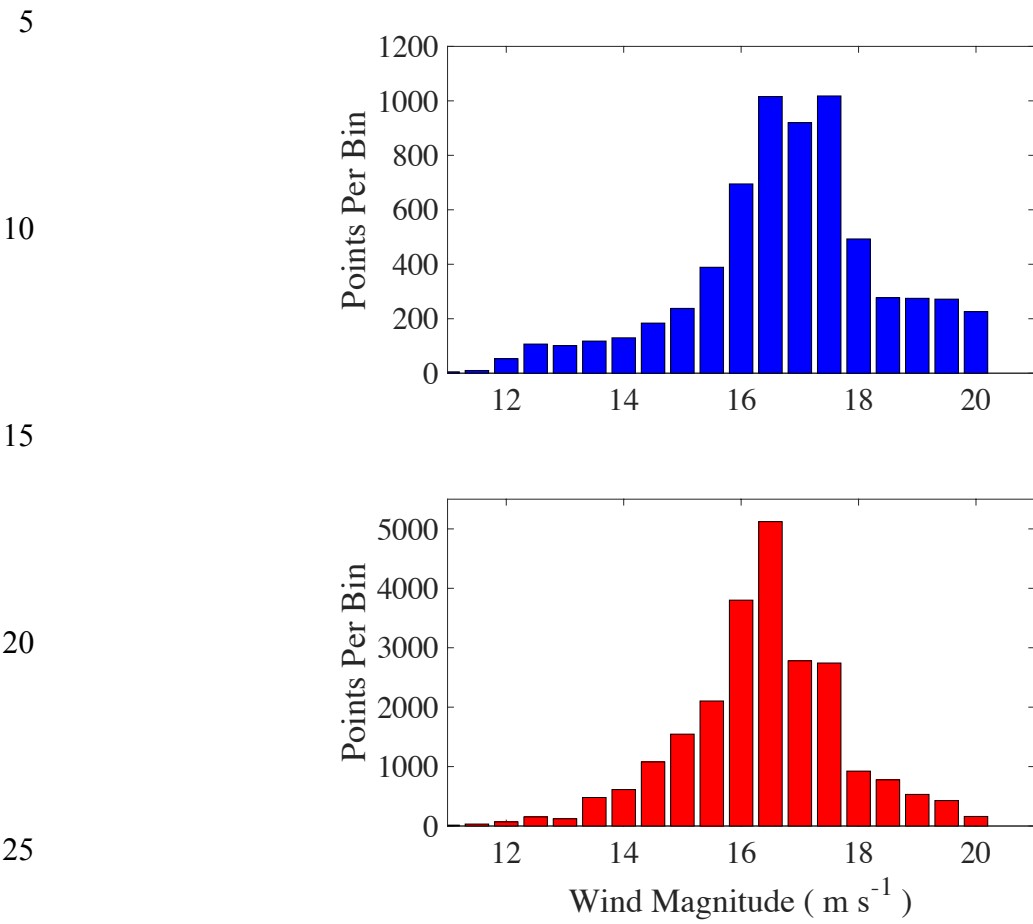

**Figure 13: Number of undrogued (top) and drogued (bottom) drifter positions where the velocity deconstruction was performed, binned by wind velocity magnitude on 0.5 m s$^{-1}$ intervals. Point velocity measurements calculated from drifter trajectories that do not coincide spatially with available data in the pre-existing circulation velocity fields are not included.**

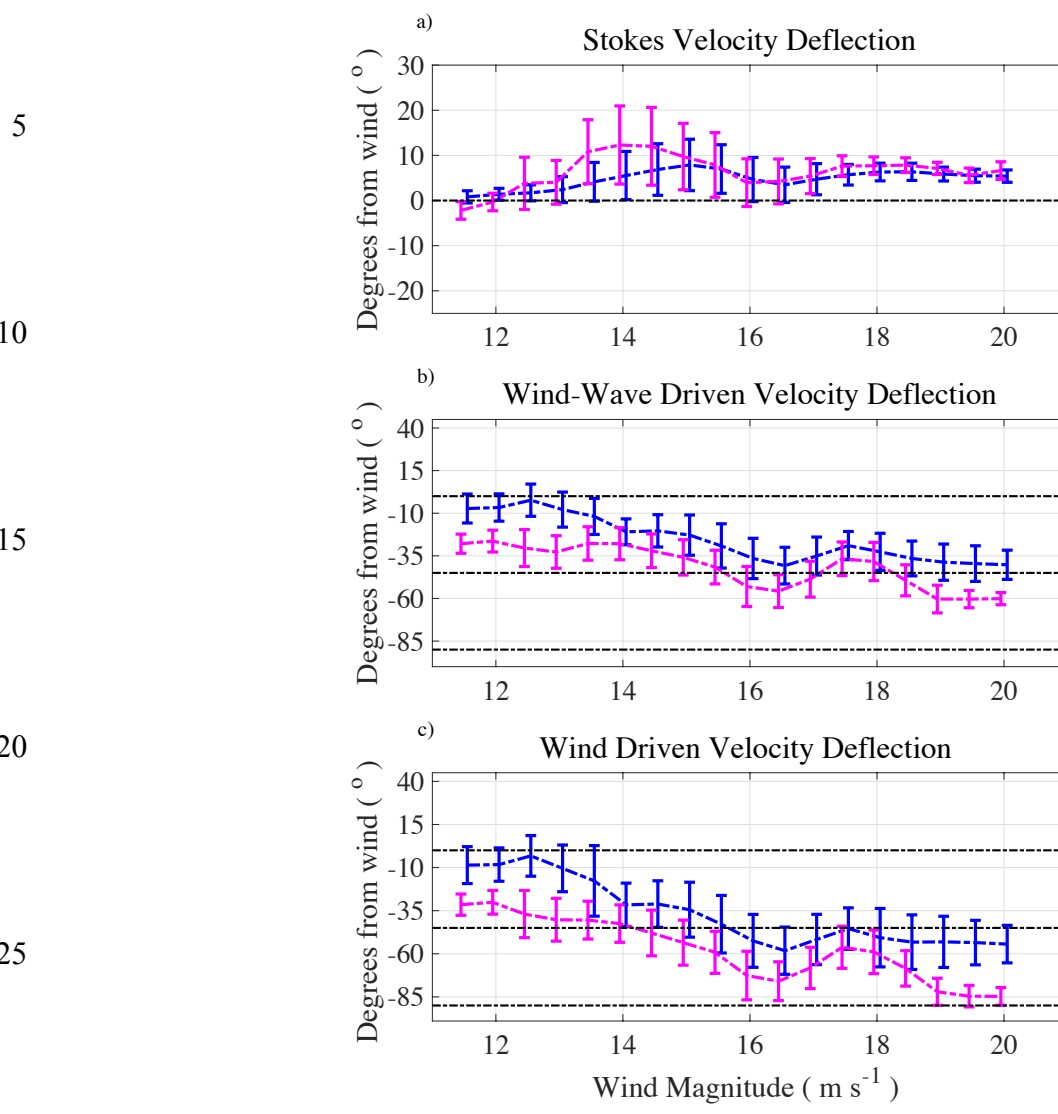

**Figure 14: Average deflection angles at drifter locations during periods of high wind binned by UWIN-CM wind magnitudes on 0.5 m s⁻¹ intervals. Horizontal dotted lines denote 0°, 45°, and 90° to the right of the wind. Blue lines indicate undrogued drifter associated velocities, while magenta lines indicate the same for drogued drifters. (a) UWIN-CM Stokes drift velocity direction at the surface (0 m) and at 0.4 m depth. (b) Velocity direction of drogued and undrogued drifters after the subtraction of the pre-existing circulation estimate. (c) Velocity direction of the purely wind-driven component of the drifter velocities. All error bars show +/- one standard deviation within each bin. Data in wind bins lower than 11.5 m s⁻¹ were omitted due to lack of data points and large standard deviations.**

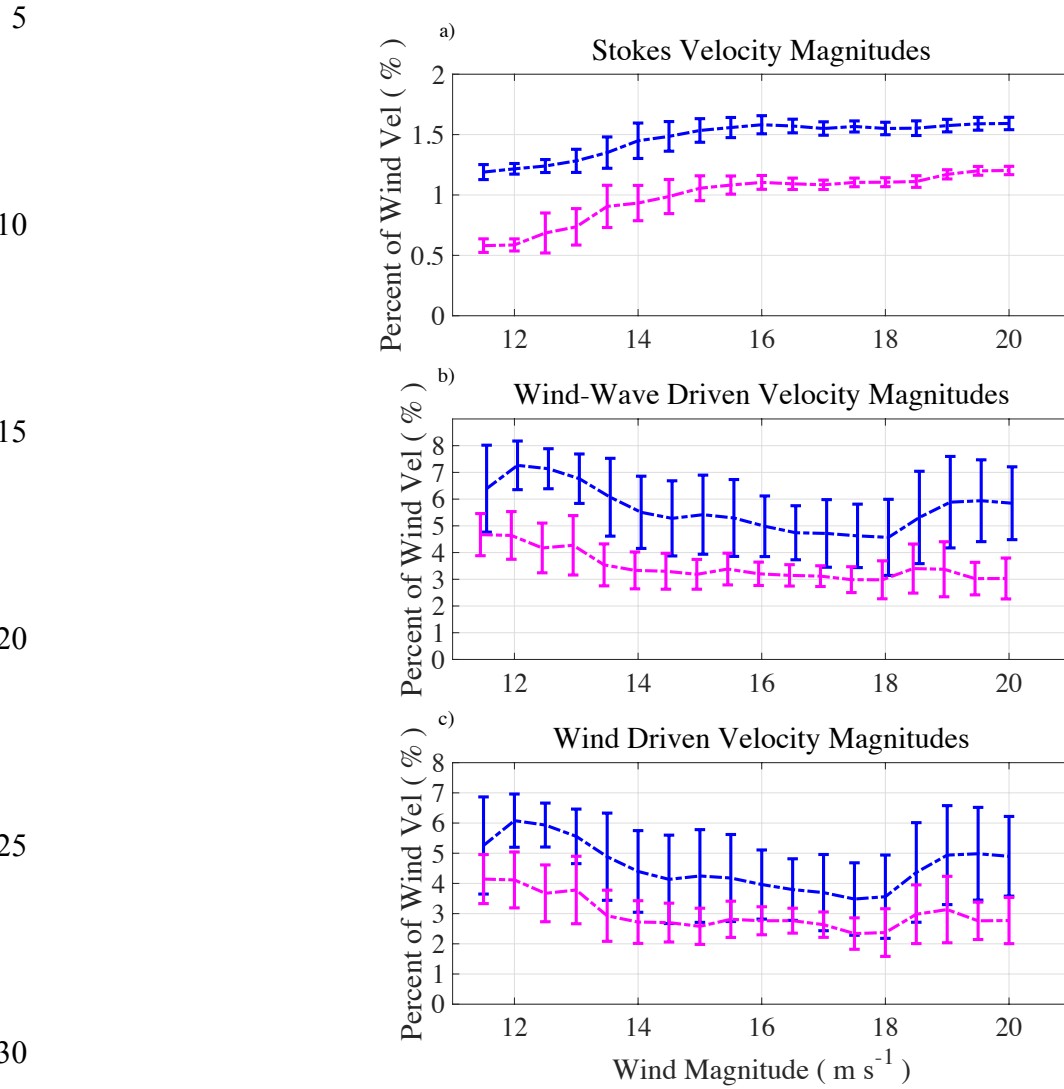

**Figure 15: Average velocity magnitudes at drifter locations during periods of high wind binned by UWIN-CM wind magnitude on 0.5 m s⁻¹ intervals. Blue lines indicate undrogued drifter associated velocities, while magenta lines indicate the same for drogued drifters. (a) UWIN-CM Stokes drift velocity magnitude at the surface (0 m) and at 0.4 m depth. (b) Velocity magnitude of drogued and undrogued drifters after the subtraction of the pre-existing circulation estimate. (c) Velocity magnitude of the purely wind-driven component of the drifter velocities. All error bars show +/- one standard deviation within each bin. Data bins omitted in Figure 14 are also excluded.**

