# Peer review of "Vertical Structure of Ocean Surface Currents Under High Winds from Massive Arrays of Drifters"

_Ocean Science, 2019_

## Referee Comment (RC1) · Anonymous Referee #1 · 30 Apr 2019

The paper addresses the magnitude and direction of surface currents under the presence of strong wind forcing, assessed using drouged and undrouged surface drifters. Given the large uncertainties of today's theoretical and numerical models with respect to surface currents and their vertical variation, this is an urgent topic. The presented studies uses an extensive data set in a dynamic interesting setting, and provides results that can serve as examplary quantification of surface currents under strong wind forcing. The work therefore has a potential to become a useful reference for such dynamics. There are a few shortcomings in the analysis and the presentation of the work, requiring major revision before the work should be published.

A potential flaw in the analysis lies in the determination of a pre-existing circulation field that is supposed to be constant throughout the remainder of the experiment. As

justification, the authors refer to very general descriptions of circulation in the region and to a sudden increase in wind-speed, but Fig. 2 rather shows that there is a steady increase of winds over one day. In addition, any pre-existing current is subject to further development and changes typically occur withing one inertial period. For example, previous wind or buoyancy forcing events could have set up near-inertial oscillations that continue to change throughout the next wind event. This has to be addressed in a revised paper, potentially involving another method to identify background currents that are allowed to be time-variable. A snapshot at one arbitrary time step is not convincing enough, even if subtraction of that current field helps to collapse scatter in wind- versus current comparison.

A second shortcoming of the study that is straightforward to address is the structure and organisation of the paper. Introductory material, methods, result and discussion is often mixed up in the paper and it would help to re-organize or re-write some parts. For example, A description of the applied method is given in the first paragraph of the introduction, which should be given in section 3 or in the abstract. Some background information and literature reviews are given in later parts of the paper, that should have better fitted in the introduction, e.g. page 5 line 5-24 and page 13 line 19-31. In the second-last paragraph of the introduction, results are given. A major part of the results is provided in the discussion section. These jumps in the paper make it very difficult to read. At the same time, a lot of needless redundandcies are made, suggesting that the paper could be shortened and sharpened to some extent.

Specific comments:

- A motivation for the study should be given in the abstract. I first realized what the overall motive was when I read the discussion.

- Page 2 line 17-21. Beside CODE and SVP drifters, undrouged SVP's and co-called iSpheres, and bamboo plates have been used to measure currents at the very surface.

- Page 2 line 22: It is quite possible to measure surface currents using ADCPs, for

example using Nortek's Signature 1000 ADCP.

- Page 3 line 9-1 and page 15, line 12-15: The authors claim novelty with regard to estimating vertical shear between the 0-5cm and 0-60cm layers. Notice that the difference between undrogued drifters and drifters with drogue at ∼1m depth has been quantified in Röhrs, J. & Christensen, K. H. Drift in the uppermost part of the ocean. Geophy. Res. Lett. 42, 1–8 (2015), and in Morey, S. L., Wienders, N., Dukhovskoy, D. S. & Bourassa, M. A. Measurement Characteristics of Near-Surface Currents from Ultra-Thin Drifters, Drogued Drifters, and HF Radar. Remote Sens.-Basel 10, 1633 (2018).

- pape 9, line 1-8 should be re-written

- Possible windage of the used surface drifters should be addressed.

---

## Referee Comment (RC2) · Anonymous Referee #2 · 2 May 2019

The paper by Lodise et al concerns the behavior of undrogued and drogued so-called CARTHE drifters. I agree with Ref #1 that lack of structure is a major problem in this paper, it is surprisingly hard to follow the authors as they describe their methods and present their results. A few recent papers are lacking, although Ref #1 mentions that as well.

What concerns me the most is the mix of tools used here in combination with a lack of uncertainty estimates: A coupled atmosphere-wave-ocean modeling system is used to provide the physical quantities needed to analyse the drift, but instead of using the ocean model results for the vaguely defined "pre-existing" circulation, the drifter trajectories are used to provide ocean circulation estimates that are later used in the analysis of the drifters themselves. Even though one can argue that the circulation estimates

are independent of the drifter observations from the "high wind" periods later on, I still wonder how representative this "pre-existing circulation" is and feel uncomfortable about this circular use of the drifter data. Why discard the ocean model component entirely? Isn't the upper layer in the model even representative of the background circulation? If not, what is the point of the ocean coupling? And surely there are other ocean models that could be used, at least for comparison to the LAVA results? The model coupling will also need to be better described: How is the coupling implemented? Are wave-dependent air-sea momentum fluxes part of the coupling (both to atmosphere and ocean) for instance? It is precisely in situations with rapidly changing weather conditions that these couplings become important (see Rohrs et al., Ocean Dyn., 2012, for estimates of wave dependent fluxes from directional wave rider data and an analysis of upper ocean drift relevant to the paper under review). Although it appears to be a good idea to split the upper ocean drift velocity into "wave", "wind" and "background" components, it really depends on the ability to delineate them, and I'm not entirely convinced in this case. The CARTHE drifters are influenced by the Stokes drift, how much of the LAVA circulation estimates still contain some Stokes drift despite making model based corrections etc.? I'm not sure what the authors should do to present a more convincing case, but I would at least need to see a much clearer presentation with more emphasis on the various potential sources of error in the estimates. Alternatives to the LAVA circulation estimates should definitely be considered (and compared to the LAVA estimates). This work would constitute a major revision of the current manuscript.

Some minor comments: Eq. (1) isn't an average, "1/h" is lacking. A short description of the typical hydrographic structure of the upper ocean in the region (mixed layer depths, buoyancy frequencies, salinity/temperature profiles etc.) would be nice. Please add some info about the spectral resolution in the wave model as well. Eq. (2) is not correct for a model with the limit at infinity, there is a cutoff frequency, and presumably a specific shape of the spectrum is assumed.

---

## Referee Comment (RC3) · Fabrice Ardhuin (Referee) · 3 May 2019

General comments: This paper presents very interesting data on near-surface drifter velocities that are interpreted in terms of near surface currents. This interpretation probably requires a clearer explanation of how the drifter velocity related to the surrounding water motion. I strongly encourage the authors to clarify this and resubmit their paper.

Specific comments:

1. The author write about "surface current" when they actually mean "drifter motion". The distinction is important as the drifter without the drogue will move due to the direct effect of the wind and of the waves, in addition to that of the current. In Novelli et

al. (2017), the undrogued drifter moves at a speed that is significantly larger than the surface current (their figures 10 and 11), of the order of 10 cm/s for 10 m/s wind. However, it is unclear how this difference scales to the open ocean with very different wave ages and vertical mixing: this cannot scale with the Stokes drift (the Stokes drift in the lab is under 0.5% of the wind speed... ). It is very unclear how the difference between water and drifter motion is estimate or corrected.

2. The idea of a "purely wind-driven current" should be clarified, in particular how the time varying wind produces a time-varying current, including a phase shift in time. On page 9, line 9, I guess there is some wind influence already in the "pre-existing regional circulation, u_rc"

3. There is not a single mention of density, temperature or salinity in the paper. It is expected that the surface response to the wind is very sensitive to the stratification (slippery layers, e.g. Kudryavtsev et al. JPO 1990). So that the present data is impossible to interpret without that information in the context of the wider litterature.

4. Numerical models or parameterizations of waves primarily design to get wave heights can disagree a lot on the short wave purely wind-driven current components that contribute to the Stokes drift (e.g. Peureux et al. 2008). Hence it would bee good to show a specific model validation on the wave spectrum in the 1 m to 40 m wavelengths regime that dominates the Stokes drift.

Technical corrections:

- Page 1: line 28: replace CARHTE with CARTHE

- in the paragraph "Observational data that captures the vertical shear within the first meter of wind-driven surface currents is very limited in the real ocean as well" the authors could reference some important work (Santala & Terray 1992)

- Page 2, line 28: "twice as fast as the average current over the first 1m and four times as fast" is misleading as a casual reader could think that over a 2 m/s Gulf Stream he

would also have a 2 m/s wind shear. Please give a velocity difference in cm/s and / or scale it with the wind speed. Please also note that these shears should be mixed by wave breaking and should thus be much smaller in the open ocean than in the lab or in coastal areas / weak winds. As a result, lab studies are largely irrelevant for the open ocean. In that respect, Sutherland et al. (2016) is a relevant reference.

- Page 2: line 31. Classical Ekman theory stricto sensu (in particular the 45° !) does not apply to the real ocean. Please consider at least realistic mixing (Madsen 1977 or Rascle et al. 2009 are better).

Page 2: line 24: 0.5 m is optimistic,

Page 3, line 5: "anywhere from 0.4 % to 5 %" is not a scientific statement. The uncertainty is much less than this range, as most of the variability in horizontally homogeneous conditions is known function of the wind speed and stratification (Ardhuin et al. 2009). Besides, I did not find in Berta et al. a clear numbers on a "wind-only' component.

Page 4, line 31: please replace "current" with "drifter velocity"

Page 5, line 9: The acceleration is not just due to Stokes drift as shown in Novelli et al. (Stokes drift at low wind is under 1% of wind speed).

Page 5: lines 16-19: I would not expect that separation changes so much the mean wind speed in the ocean over the near-surface 10 cm. The radiation stress of the short waves dissipated / reflected by the obect can be relevant, see Longuet-Higgins 1977.

Page 5: Given the very different wave age in the lab and in the field, it is not clear at all that the "velocity slip" in the lab can be scaled to the field conditions.

Page 6: Please show / give reference to proper validation of wave model in terms of Stokes drift.

Page 10 line 4-5: please be more specific and replace "wind-driven velocities" by "winddriven drifter velocities"

Page 12, line 31: "possibly the most significant" is a pretty bold comment given the history of the field (Munk 2002). I would contend that stratification is the elephant in the room here.

Page 13: Please do not use the word "current" unless you are properly explaining how you go from drifter velocity to water velocity.

Page 13, line 22: please clarify if that includes the Stokes drift or not. Also, it should be important to discuss the effect of proximity to coast as the wind-driven current are rectified by the shoreline in many datasets.

Page 15, line 1: "The momentum input from large breaking waves into the surface currents" what about rather, "the momentum input and surfing behaviour of undrogued drifters in large breaking waves"

Page 15 line 11: " twice as fast " does not make much sense, please provide some scale (wind, Stokes drift ...) you do not expect to go twice as fast in the top meter above a 2 m/s Gulf Stream. Also please discuss stratification.

References: Rascle, N., & Ardhuin, F. (2009). Drift and mixing under the ocean surface revisited: Stratified conditions and model-data comparisons. Journal of Geophysical Research, 114(C2). doi:10.1029/2007jc004466

Santala, M. J., and E. A. Terray (1992), A technique for making unbiased estimates of current shear from a wave-follower, Deep Sea Res., Part A, 39, 607 – 622.

Sutherland et al. JPO 2016.

@ARTICLE{Sutherland&al.2016, author = "Graig Sutherland and Louis Mari{\'e} and Gilles Reverdin and Kai H. Christensen and G{\"o}ran Brostr{\"o}m and Brian Ward", title = "Enhanced Turbulence Associated with the Diurnal Jet in the Ocean Surface Boundary Layer", journal = JPO, volume = 46, pages = "3051–3067", year = 2016,

doi=" 10.1175/JPO-D-15-0172.1", keyword="surface velocity", where="PDF", }

Kudryavtsev, V. N., & Soloviev, A. V. (1990). Slippery Near-Surface Layer of the Ocean Arising Due to Daytime Solar Heating. Journal of Physical Oceanography, 20(5), 617–628. doi:10.1175/1520-0485(1990)020<0617:snslot>2.0.co;2

@ARTICLE{Longuet-Higgins1977, author = "Michael S. Longuet-Higgins", title = "The mean forces exerted by waves on floating or submerged bodies with applications to sand bars and wave power machines", journal = PRSLA, volume = 352, pages = "463–480", year = 1977, where="PDF", KEYWORDS={wave action;wave-mean flow interactions}, }

Munk, W., 2002: The Evolution of Physical Oceanography in the Last Hundred Years, Oceanography, 15(1), doi: 10.5670/oceanog.2002.45

@ARTICLE{Peureux&al.2018, author = "Charles Peureux and Alvise Benetazzo and Fabrice Ardhuin", title = "Note on the directional properties of meter-scale gravity waves", journal = "Ocean Science", volume=14, pages = "41–52", year = 2018, KEYWORDS={stereo video,sho0.0270rt wave spectrum}, doi="10.5194/os-14-41-2018", }

---

## Author Comment (AC1) · 31 May 2019

**Response to Anonymous Referee #1**

The paper addresses the magnitude and direction of surface currents under the presence of strong wind forcing, assessed using drogued and undrogued surface drifters. Given the large uncertainties of today's theoretical and numerical models with respect to surface currents and their vertical variation, this is an urgent topic. The presented study uses an extensive data set in a dynamic interesting setting, and provides results that can serve as examplary quantification of surface currents under strong wind forcing. The work therefore has a potential to become a useful reference for such dynamics. There are a few shortcomings in the analysis and the presentation of the work, requiring major revision before the work should be published.

*We would like to thank Referee #1 for his or her constructive feedback on the manuscript*

A potential flaw in the analysis lies in the determination of a pre-existing circulation field that is supposed to be constant throughout the remainder of the experiment. As justification, the authors refer to very general descriptions of circulation in the region and to a sudden increase in wind-speed, but Fig. 2 rather shows that there is a steady increase of winds over one day. In addition, any pre-existing current is subject to further development and changes typically occur within one inertial period. For example, previous wind or buoyancy forcing events could have set up near-inertial oscillations that continue to change throughout the next wind event. This has to be addressed in a revised paper, potentially involving another method to identify background currents that are allowed to be time-variable. A snapshot at one arbitrary time step is not convincing enough, even if subtraction of that current field helps to collapse scatter in wind-versus current comparison.

> *We have adapted the figure showing the LAVA estimates (fig. 3 in original version) into 2 new figures (Figs. 6-7) showing the addition of AVISO SST maps plotted beneath the velocity fields. SST maps from AVISO are created every 24-hours centered on the $0^{th}$ hour of each day. Two SST maps are shown with identical LAVA estimates to show that during and even after the window of analysis under high winds, the structures seen in the velocity fields appear consistent with AVISO observations throughout each high wind event. The SST maps show structures that are qualitatively similar to the velocity fields created with LAVA, even over the 24-hour product created after our analysis period.*
>
> *Descriptions of these figures (6-7) and their implications can be found on:*
> *Page 10, lines 12-28*
> *Page 12, lines 9-23*
> *Page 16, lines 17-24*

A second shortcoming of the study that is straightforward to address is the structure and organisation of the paper. Introductory material, methods, result and discussion is often mixed up

in the paper and it would help to re-organize or re-write some parts. For example, A description of the applied method is given in the first paragraph of the introduction, which should be given in section 3 or in the abstract. Some background information and literature reviews are given in later parts of the paper, that should have better fitted in the introduction, e.g. page 5 line 5-24 and page 13 line 19-31. In the second-last paragraph of the introduction, results are given. A major part of the results is provided in the discussion section. These jumps in the paper make it very difficult to read. At the same time, a lot of needless redundandcies are made, suggesting that the paper could be shortened and sharpened to some extent.

*Some of the paper has been reorganized based on the reviewer's suggestions.*

*The first paragraph of the introduction has been combined with a later paragraph of the introduction in order to reduce redundancy. (Page 3, lines 15-26).*

*The background information on Page 5 line 5-24, of the previous version of the paper, pertain specifically to the drifters used in this study, therefore the authors feel it appropriate to leave this description under the section titled 2.1 CARTHE drifter.*

*The authors also feel it appropriate to leave the description on what was formerly Page 13 line 19-31 in the previous version of the paper as is, as it acts a review of the significant body of work available on the topic. This lines act to help the reader remember specific details from the studies, we wish to reiterate in comparison to our results.*

*In the second to last paragraph no change was made. A brief description of the outcome from the subtraction of the regional circulation estimate, as well as a reference to a previous study, is presented to act as motivation for the method proposed.*

*Two paragraphs from the discussion section were moved to the results section, as requested by reviewer. They now reside on Pages 13, line 28 – page 14, line 11. As a result of this edit, we also rewrote part of the discussion, seen on page 15, lines 1-6.*

*Attempts were made to reduce redundancies in the discussion section, with the omission of Page 12, lines 2-6, in the original version.*

Specific comments:

- A motivation for the study should be given in the abstract. I first realized what the overall motive was when I read the discussion.

*We have added more on motivation to the abstract.*

*Page 1. Lines 9-10.*

- Page 2 line 17-21. Beside CODE and SVP drifters, undrouged SVP's and co-called iSpheres, and bamboo plates have been used to measure currents at the very surface.

*These other types of drifters are mentioned elsewhere in the paper. This point was made to show other studies have missed the vertical shear in the top 1m.*

- Page 2 line 22: It is quite possible to measure surface currents using ADCPs, for example using Nortek's Signature 1000 ADCP.

*To our knowledge it is very difficult to measure surface currents at shallow depths, less than ~0.5m, in the presence of surface gravity waves due to side lobe contamination caused by the surface wave motion (Cole and Symonds 2015).*

*Cole, R., & Symonds, D.: A 25 year collaboration using ADCPs. In 2015 IEEE/OES 11th Current, Waves and Turbulence Measurement Workshop, St. Petersburg, FL, USA, March 2015, 1–10, 2015.*

- Page 3 line 9-1 and page 15, line 12-15: The authors claim novelty with regard to estimating vertical shear between the 0-5cm and 0-60cm layers. Notice that the difference between undrogued drifters and drifters with drogue at ~1m depth has been quantified in Röhrs, J. & Christensen, K. H. Drift in the uppermost part of the ocean. Geophy. Res. Lett. 42, 1–8 (2015), and in Morey, S. L., Wienders, N., Dukhovskoy, D. S. & Bourassa, M. A. Measurement Characteristics of Near-Surface Currents from Ultra-Thin Drifters, Drogued Drifters, and HF Radar. Remote Sens.-Basel 10, 1633 (2018).

*These references have been noted and added to the manuscript and claims of novelty have been revised.*

*Page 2, lines 24-29*

*Page 3, lines 3-10 and lines 23-26*

*Page 15, lines 24-27*

*Page 16, lines 3-9*

*Page 17, lines 18-31*

*Page 18, line 23*

- pape 9, line 1-8 should be re-written

*lines have been rewritten in addition to extra explanation*

*Page 10, line2 – Page 11, line 6.*

- Possible windage of the used surface drifters should be addressed.

*The exact windage on the drifters, especially for the undrogued drifter in the real ocean in not known to an extent in which we can apply a meaning correction. We have tried to be as transparent and straight forward to the extent to which possible errors pertaining to windage are expected.*

*Page 17, lines 24-26 and lines 10-13*

*Page 6, lines 7-10.*

---

## Author Comment (AC2) · 31 May 2019

> *We would also like to thank Referee #2 for his or her time and insight to the current manuscript*

The paper by Lodise et al concerns the behavior of undrogued and drogued so-called CARTHE drifters. I agree with Ref #1 that lack of structure is a major problem in this paper, it is surprisingly hard to follow the authors as they describe their methods and present their results. A few recent papers are lacking, although Ref #1 mentions that as well.

> *References mentioned by Ref 1 have been added.*

> *Page 2, lines 24-29*

> *Page 3, lines 3-10*

> *Page 15, lines 24-27*

> *Page 16, lines 3-9*

> *Page 18, line 23*

What concerns me the most is the mix of tools used here in combination with a lack of uncertainty estimates: A coupled atmosphere-wave-ocean modeling system is used to provide the physical quantities needed to analyse the drift, but instead of using the ocean model results for the vaguely defined "pre-existing" circulation, the drifter trajectories are used to provide ocean circulation estimates that are later used in the analysis of the drifters themselves. Even though one can argue that the circulation estimates are independent of the drifter observations from the "high wind" periods later on, I still wonder how representative this "pre-existing circulation" is and feel uncomfortable about this circular use of the drifter data. Why discard the ocean model component entirely? Isn't the upper layer in the model even representative of the background circulation? If not, what is the point of the ocean coupling? And surely there are other ocean models that could be used, at least for comparison to the LAVA results?

> *When compared to the LAVA velocity fields created using the CARTHE drifters, the surface velocity fields from the ocean circulation component of the UWIN-CM, qualitatively, do not agree (shown in the figure here). At the scales sampled by the drifters in this study, it becomes very difficult to model ocean circulation due to the lack of data to serve as initial conditions, especially in an area rich in frontal activity such as this. This has been tested in a number of previous papers (e.g. Chang et al., 2011; Taillandier et al., 2008), where drifter based LAVA fields were compared to model*

outputs and used to improve model performances. The drifters themselves would serve as the best validation for any ocean circulation model, therefore using the drifter data itself to come up with estimates of the regional circulation is expected to be more accurate.

The model runs were performed during the LASER experiment not only to be used as a data set for later analysis, but also for operational procedures of the experiment itself. Even though the ocean circulation model doesn't seem to be representative of reality in this region, the wind and wave data from the other 2 model components perform well when compared to observational data.

*LAVA Drogued Velocity Field*                    *UWIN-CM Ocean Circulation at 0.5 m*

[Figure]

*REFERENCES:*

*Chang, Y., D. Hammond, A. Haza, P. Hogan, H.S. Huntley, A.D. Kirwan, Jr, B.L. Lipphardt, Jr., V. Taillandier, A. Griffa, T.M. Ozgokmen, 2011. Enhanced Estimation of Sonobuoy Trajectories by Velocity Reconstruction With Near-Surface Drifters. Oc. Modelling, 36, 179-197*

*Taillandier, A. Griffa, P.M. Poulain, R. Signell, J. Chiggiato, S. Carniel, 2008. Variational analysis of drifter positions and model outputs for the reconstruction of surface currents in the Central Adriatic during fall 2002. J. Geophys. Res., 113, C04004, doi.1029/2007/JC004148*

The model coupling will also need to be better described: How is the coupling implemented? Are wave-dependent air-sea momentum fluxes part of the coupling (both to atmosphere and ocean) for instance? It is precisely in situations with rapidly changing weather conditions that these couplings become important (see Rohrs et al., Ocean Dyn., 2012, for estimates of wave dependent fluxes from directional wave rider data and an analysis of upper ocean drift relevant to the paper under review).

> *The coupling implementation has been described in the manuscript.*

> *Page 7, lines 1-6*

Although it appears to be a good idea to split the upper ocean drift velocity into "wave", "wind" and "background" components, it really depends on the ability to delineate them, and I'm not entirely convinced in this case. The CARTHE drifters are influenced by the Stokes drift, how much of the LAVA circulation estimates still contain some Stokes drift despite making model based corrections etc.?

> *The average Stokes drift during the selected hours over which the pre-existing circulation has been estimated using LAVA is 0.02 m/s and 0.04 m/s for the drogued and undrogued drifter cases, respectively. Due to the velocity spreading and averaging performed by LAVA, the uncertainty which arises from the Stokes drift component is small during these time periods.*

> *Page 10, lines 21-28*

I'm not sure what the authors should do to present a more convincing case, but I would at least need to see a much clearer presentation with more emphasis on the various potential sources of error in the estimates. Alternatives to the LAVA circulation estimates should definitely be

considered (and compared to the LAVA estimates). This work would constitute a major revision of the current manuscript.

> *In addition, we have added 2 new figures (Figs 6 and 7) to make a more convincing case that the LAVA velocity fields are fair estimates of the pre-existing circulation by plotting the velocity fields on top of 2 consecutive 24-hour averages of SST data from AVISO. The first 24-hour averages for the SST span the time periods over which the high wind analysis windows are defined, where the second 24-hour periods, span the 24-hours following the high wind event. The structures seen in the SST fields line up very well with the LAVA velocity estimates of the pre-existing circulation and the second 24-hour period of SST shows qualitatively how much these structured changed during the high wind event. With these figures, we've hoped to have further convinced the reader that the estimates for the pre-existing regional circulation are adequate for our velocity deconstruction.*

> *Descriptions of these figures (6-7) and their implications can be found on:*
> *Page 10, lines 12-28*
> *Page 12, lines 9-23*
> *Page 16, lines 17-24*

Some minor comments:

Eq. (1) isn't an average, "1/h" is lacking.

> *You are correct. Thank you, the change has been made.*

> *Page 3, line 31*

A short description of the typical hydrographic structure of the upper ocean in the region (mixed layer depths, buoyancy frequencies, salinity/temperature profiles etc.) would be nice.

> *A short description as well as a new figure (Fig. 2) has been added to describe the typical stratification and mixed layer depth in terms of temperature, salinity and potential density, in the presence of a typical frontal feature observed during the experiment.*

> *Page 7, lines 22-31.*

Please add some info about the spectral resolution in the wave model as well. Eq. (2) is not correct for a model with the limit at infinity, there is a cutoff frequency, and presumably a specific shape of the spectrum is assumed.

> *Proper limits for the wavenumber have been assigned in Eq. (2), and concerns about the*

*spectral resolution have been addressed.*

*Page 6, lines 21-25*

---

## Author Comment (AC3) · 31 May 2019

General comments: This paper presents very interesting data on near-surface drifter velocities that are interpreted in terms of near surface currents. This interpretation probably requires a clearer explanation of how the drifter velocity related to the surrounding water motion. I strongly encourage the authors to clarify this and resubmit their paper.

> *We would like to thank Dr. Ardhuin for his constructive review of the current work.*

Specific comments:

1. The author write about "surface current" when they actually mean "drifter motion". The distinction is important as the drifter without the drogue will move due to the direct effect of the wind and of the waves, in addition to that of the current. In Novelli et al. (2017), the undrogued drifter moves at a speed that is significantly larger than the surface current (their figures 10 and 11), of the order of 10 cm/s for 10 m/s wind. However, it is unclear how this difference scales to the open ocean with very different wave ages and vertical mixing: this cannot scale with the Stokes drift (the Stokes drift in the lab is under 0.5% of the wind speed... ). It is very unclear how the difference between water and drifter motion is estimate or corrected.

> *We agree that it is very difficult to determine how the wave action and especially the wind will affect the movement of the undrogued drifters during the conditions being analyzed here. For this reason, we do not make any correction of the drifter velocity, but choose to leave the manner in which we describe the "surface currents" unchanged. We believe it makes sense to describe the drifter velocities as the average current over the draft depth of the drifter, given the definition portrayed in Eq. (1). We have tried to be more precise in our description of the surface current estimate and the errors associated with the drifter movement due to velocity slip.*
>
> *Page 6, lines 7-10.*
>
> *Page 17, lines 24-26 and lines 10-13*

2. The idea of a "purely wind-driven current" should be clarified, in particular how the time varying wind produces a time-varying current, including a phase shift in time. On page 9, line 9, I guess there is some wind influence already in the "pre-existing regional circulation, u_rc"

*Effect of wind and waves on drifters during the low wind period over which the regional circulation estimates have been described in section 3.2, along with new Figures showing SST for validation (Figs. 6-7).*

*Page 10, lines 21-29*

*Response time of surface currents has also been addressed with supporting literature.*

*Page 11, lines 1-5*

3. There is not a single mention of density, temperature or salinity in the paper. It is expected that the surface response to the wind is very sensitive to the stratification (slippery layers, e.g. Kudryavtsev et al. JPO 1990). So that the present data is impossible to interpret without that information in the context of the wider literature.

*A new figure (Fig. 2) has been added to show a typical transects of salinity, temperature, and potential density across a frontal zone in the region measured during the experiment, however not during one of the high wind periods. Fronts like this one were frequently measured during this experiment showing transects very similar to the one shown in Fig. 2.*

*Page 7, lines 22-31.*

*Effects of stratification have also been referenced in Introduction*

*Page 3, lines 10-14.*

4. Numerical models or parameterizations of waves primarily design to get wave heights can disagree a lot on the short wave purely wind-driven current components that contribute to the Stokes drift (e.g. Peureux et al. 2008). Hence it would be good to show a specific model validation on the wave spectrum in the 1 m to 40 m wavelengths regime that dominates the Stokes drift.

*We've added a new Figure (Fig. 4) to the manuscript to provide wave validation with available observations from the model during one of the high wind events, showing significant wave height, mean wave direction, wavelength, and mean wave period. With only hourly averages from NDBC buoy 42040 to compare to the model too, we felt this figure was more appropriate than trying to calculate wave spectrum from hourly averaged data.*

*Page 8, lines 11-19*

Technical corrections:

- Page 1: line 28: replace CARHTE with CARTHE

 *We have replaced this. Thank you*

- in the paragraph "Observational data that captures the vertical shear within the first meter of wind-driven surface currents is very limited in the real ocean as well" the authors could reference some important work (Santala & Terray 1992)

 *We have cited Santala and Terray (1992)*

 *Page 2, lines 13-15*

- Page 2, line 28: "twice as fast as the average current over the first 1m and four times as fast" is misleading as a casual reader could think that over a 2 m/s Gulf Stream he would also have a 2 m/s wind shear. Please give a velocity difference in cm/s and / or scale it with the wind speed. Please also note that these shears should be mixed by wave breaking and should thus be much smaller in the open ocean than in the lab or in coastal areas / weak winds. As a result, lab studies are largely irrelevant for the open ocean. In that respect, Sutherland et al. (2016) is a relevant reference.

 *A reference velocity has been provided in regard to this description of the vertical velocity shear. A note about breaking waves and mixing in regard to decreased velocity shear has also been referenced, citing Sutherland et al. 2016*

 *Page 2, line 24*
 *Page 17, lines 22-26*

Page 2: line 31. Classical Ekman theory stricto sensu (in particular the 45◦ !) does not apply to the real ocean. Please consider at least realistic mixing (Madsen 1977 or Rascle et al. 2009 are better).

 *Rascle et al. 2009 has been described and cited.*

 *Page 3, Lines 13-14*

Page 2: line 24: 0.5 m is optimistic

 *0.5 m was the value from references listed.*

Page 3, line 5: "anywhere from 0.4 % to 5 %" is not a scientific statement. The uncertainty is

much less than this range, as most of the variability in horizontally homogeneous conditions is known function of the wind speed and stratification (Ardhuin et al. 2009). Besides, I did not find in Berta et al. a clear number on a "wind-only' component.

*This range of findings for the wind-driven current is based off the range of results reported by the different studies cited. The lines were edited for clarity. Berta et al. 2018 states their ageostrophic component, which is dominated by the wind, travels at ~2 % of the wind speed.*

*Page 3, lines 7-8*

Page 4, line 31: please replace "current" with "drifter velocity"

*We would like to keep the present convention of referring to the drifter velocities as measurements of the surface currents themselves, with potential errors described.*

Page 5, line 9: The acceleration is not just due to Stokes drift as shown in Novelli et al. (Stokes drift at low wind is under 1% of wind speed).

*We were referring to lab experiments where there was no wind influence. This point has been clarified in the text.*

*Page 5, line 22*

Page 5: lines 16-19: I would not expect that separation changes so much the mean wind speed in the ocean over the near-surface 10 cm. The radiation stress of the short waves dissipated / reflected by the obect can be relevant, see Longuet-Higgins 1977.

*The mechanics of radiation stress on the effect of the drifters is non-trival to diagnose, and is thus beyond the scope of this paper.*

Page 5: Given the very different wave age in the lab and in the field, it is not clear at all that the "velocity slip" in the lab can be scaled to the field conditions.

*We agree it probably cannot, which is why we choose not to perform a velocity correction to the measured drifter velocities.*

Page 6: Please show / give reference to proper validation of wave model in terms of Stokes drift.

*Wave model validation provided in Fig. 4*

*Page 8, lines 11-19*

Page 10 line 4-5: please be more specific and replace "wind-driven velocities" by "wind driven drifter velocities"

*Again, we would like to keep the present convention of referring to the drifter velocities as measurements of the surface currents themselves, with potential errors described.*

Page 12, line 31: "possibly the most significant" is a pretty bold comment given the history of the field (Munk 2002). I would contend that stratification is the elephant in the room here.

*Paragraphs have been edited with the stratification mentioned as well.*

*Page 15, line 15.*

Page 13: Please do not use the word "current" unless you are properly explaining how you go from drifter velocity to water velocity.

*We refer to the drifter velocities as measurements of the surface currents themselves, with potential errors described.*

Page 13, line 22: please clarify if that includes the Stokes drift or not. Also, it should be important to discuss the effect of proximity to coast as the wind-driven current are rectified by the shoreline in many datasets.

*Clarification of the inclusion of stokes drift and effect of shoreline rectification has been made.*

*Page 16, lines 4-9*

Page 15, line 1: "The momentum input from large breaking waves into the surface currents" what about rather, "the momentum input and surfing behaviour of undrogued drifters in large breaking waves"

*Statement has been changed to include this phenomenon.*

*Page 10-13*

Page 15 line 11: " twice as fast " does not make much sense, please provide some scale (wind, Stokes drift ...) you do not expect to go twice as fast in the top meter above a 2 m/s Gulf Stream. Also please discuss stratification.

*Absolutely velocity measurements have been provided.*

*Page 17, lines 22-25*

---

## Author Comment (AC4) · 31 May 2019

RC: The paper addresses the magnitude and direction of surface currents under the presence of strong wind forcing, assessed using drogued and undrogued surface drifters. Given the large uncertainties of today's theoretical and numerical models with respect to surface currents and their vertical variation, this is an urgent topic. The presented study uses an extensive data set in a dynamic interesting setting, and provides results that can serve as examplary quantification of surface currents under strong wind forcing. The work therefore has a potential to become a useful reference for such dynamics. There are a few shortcomings in the analysis and the presentation of the

work, requiring major revision before the work should be published.

AC: We would like to thank Referee #1 for his or her constructive feedback on the manuscript

RC: A potential flaw in the analysis lies in the determination of a pre-existing circulation field that is supposed to be constant throughout the remainder of the experiment. As justification, the authors refer to very general descriptions of circulation in the region and to a sudden increase in wind-speed, but Fig. 2 rather shows that there is a steady increase of winds over one day. In addition, any pre-existing current is subject to further development and changes typically occur within one inertial period. For example, previous wind or buoyancy forcing events could have set up near-inertial oscillations that continue to change throughout the next wind event. This has to be addressed in a revised paper, potentially involving another method to identify background currents that are allowed to be time-variable. A snapshot at one arbitrary time step is not convincing enough, even if subtraction of that current field helps to collapse scatter in wind- versus current comparison.

AC: We have adapted the figure showing the LAVA estimates (fig. 3 in original version) into 2 new figures (Figs. 6-7) showing the addition of AVISO SST maps plotted beneath the velocity fields. SST maps from AVISO are created every 24-hours centered on the 0th hour of each day. Two SST maps are shown with identical LAVA estimates to show that during and even after the window of analysis under high winds, the structures seen in the velocity fields appear consistent with AVISO observations throughout each high wind event. The SST maps show structures that are qualitatively similar to the velocity fields created with LAVA, even over the 24-hour product created after our analysis period.

Descriptions of these figures (6-7) and their implications can be found on: Page 10, lines 12-28 Page 12, lines 9-23 Page 16, lines 17-24

RC: A second shortcoming of the study that is straightforward to address is the structure and organisation of the paper. Introductory material, methods, result and discussion is often mixed up in the paper and it would help to re-organize or re-write some parts. For example, A description of the applied method is given in the first paragraph of the introduction, which should be given in section 3 or in the abstract. Some background information and literature reviews are given in later parts of the paper, that should have better fitted in the introduction, e.g. page 5 line 5-24 and page 13 line 19-31. In the second-last paragraph of the introduction, results are given. A major part of the results is provided in the discussion section. These jumps in the paper make it very difficult to read. At the same time, a lot of needless redundandcies are made, suggesting that the paper could be shortened and sharpened to some extent.

AC: Some of the paper has been reorganized based on the reviewer's suggestions.

The first paragraph of the introduction has been combined with a later paragraph of the introduction in order to reduce redundancy. (Page 3, lines 15-26).

The background information on Page 5 line 5-24, of the previous version of the paper, pertain specifically to the drifters used in this study, therefore the authors feel it appropriate to leave this description under the section titled 2.1 CARTHE drifter.

The authors also feel it appropriate to leave the description on what was formerly Page 13 line 19-31 in the previous version of the paper as is, as it acts a review of the significant body of work available on the topic. This lines act to help the reader remember specific details from the studies, we wish to reiterate in comparison to our results.

In the second to last paragraph of the introduction no change was made. A brief description of the outcome from the subtraction of the regional circulation estimate, as well as a reference to a previous study, is presented to act as motivation for the method proposed.

Two paragraphs from the discussion section were moved to the results section, as requested by reviewer. They now reside on Pages 13, line 28 – page 14, line 11. As a

result of this edit, we also rewrote part of the discussion, seen on page 15, lines 1-6.

Attempts were made to reduce redundancies in the discussion section, with the omission of Page 12, lines 2-6, in the original version.

RC: Specific comments: - A motivation for the study should be given in the abstract. I first realized what the overall motive was when I read the discussion.

AC: We have added more on motivation to the abstract. Page 1. Lines 9-10.

RC: Page 2 line 17-21. Beside CODE and SVP drifters, undrouged SVP's and co-called iSpheres, and bamboo plates have been used to measure currents at the very surface.

AC:These other types of drifters are mentioned elsewhere in the paper. This point was made to show other studies have missed the vertical shear in the top 1m.

RC: Page 2 line 22: It is quite possible to measure surface currents using ADCPs, for example using Nortek's Signature 1000 ADCP.

AC:To our knowledge it is very difficult to measure surface currents at shallow depths, less than ∼0.5m, in the presence of surface gravity waves due to side lobe contamination caused by the surface wave motion (Cole and Symonds 2015). Cole, R., & Symonds, D.: A 25 year collaboration using ADCPs. In 2015 IEEE/OES 11th Current, Waves and Turbulence Measurement Workshop, St. Petersburg, FL, USA, March 2015, 1–10, 2015.

RC: Page 3 line 9-1 and page 15, line 12-15: The authors claim novelty with regard to estimating vertical shear between the 0-5cm and 0-60cm layers. Notice that the difference between undrogued drifters and drifters with drogue at âĹij1m depth has been quantified in Röhrs, J. & Christensen, K. H. Drift in the uppermost part of the ocean. Geophy. Res. Lett. 42, 1–8 (2015), and in Morey, S. L., Wienders, N., Dukhovskoy, D. S. & Bourassa, M. A. Measurement Characteristics of Near-Surface Currents from Ultra-Thin Drifters, Drogued Drifters, and HF Radar. Remote Sens.-Basel 10, 1633

(2018).

AC:These references have been noted and added to the manuscript and claims of novelty have been revised. Page 2, lines 24-29 Page 3, lines 3-10 and lines 23-26 Page 15, lines 24-27 Page 16, lines 3-9 Page 17, lines 18-31 Page 18, line 23

RC: page 9, line 1-8 should be re-written

AC: These lines have been rewritten in addition to extra explanation Page 10, line2 – Page 11, line 6.

RC: Possible windage of the used surface drifters should be addressed.

AC:The exact windage on the drifters, especially for the undrogued drifter in the real ocean in not known to an extent in which we can apply a meaning correction. We have tried to be as transparent and straight forward to the extent to which possible errors pertaining to windage are expected. Page 17, lines 24-26 and lines 10-13 Page 6, lines 7-10.

---

## Author Comment (AC5) · 31 May 2019

Anonymous Referee #2 AC:We would also like to thank Referee #2 for his or her time and insight to the current manuscript

RC: The paper by Lodise et al concerns the behavior of undrogued and drogued so-called CARTHE drifters. I agree with Ref #1 that lack of structure is a major problem in this paper, it is surprisingly hard to follow the authors as they describe their methods and present their results. A few recent papers are lacking, although Ref #1 mentions that as well.

AC: References mentioned by Ref 1 have been added. Page 2, lines 24-29 Page 3, lines 3-10 Page 15, lines 24-27 Page 16, lines 3-9 Page 18, line 23

[Figure]

RC: What concerns me the most is the mix of tools used here in combination with a lack of uncertainty estimates: A coupled atmosphere-wave-ocean modeling system is used to provide the physical quantities needed to analyse the drift, but instead of using the ocean model results for the vaguely defined "pre-existing" circulation, the drifter trajectories are used to provide ocean circulation estimates that are later used in the analysis of the drifters themselves. Even though one can argue that the circulation estimates are independent of the drifter observations from the "high wind" periods later on, I still wonder how representative this "pre-existing circulation" is and feel uncomfortable about this circular use of the drifter data. Why discard the ocean model component entirely? Isn't the upper layer in the model even representative of the background circulation? If not, what is the point of the ocean coupling? And surely there are other ocean models that could be used, at least for comparison to the LAVA results?

AC:When compared to the LAVA velocity fields created using the CARTHE drifters, the surface velocity fields from the ocean circulation component of the UWIN-CM, qualitatively, do not agree (shown in the figures below). At the scales sampled by the drifters in this study, it becomes very difficult to model ocean circulation due to the lack of data to serve as initial conditions, especially in an area rich in frontal activity such as this. This has been tested in a number of previous papers (e.g. Chang et al., 2011; Taillandier et al., 2008), where drifter based LAVA fields were compared to model outputs and used to improve model performances. The drifters themselves would serve as the best validation for any ocean circulation model, therefore using the drifter data itself to come up with estimates of the regional circulation is expected to be more accurate. The model runs were performed during the LASER experiment not only to be used as a data set for later analysis, but also for operational procedures of the experiment itself. Even though the ocean circulation model doesn't seem to be representative of reality in this region, the wind and wave data from the other 2 model components perform well when compared to observational data.

REFERENCES: Chang, Y., D. Hammond, A. Haza, P. Hogan, H.S. Huntley, A.D. Kir-

**OSD**

wan, Jr, B.L. Lipphardt, Jr., V. Taillandier, A. Griffa, T.M. Ozgokmen, 2011. Enhanced Estimation of Sonobuoy Trajectories by Velocity Reconstruction With Near-Surface Drifters. Oc. Modelling, 36, 179-197 Taillandier, A. Griffa, P.M. Poulain, R. Signell, J. Chiggiato, S. Carniel, 2008. Variational analysis of drifter positions and model outputs for the reconstruction of surface currents in the Central Adriatic during fall 2002. J. Geophys. Res., 113, C04004, doi.1029/2007/JC004148

RC: The model coupling will also need to be better described: How is the coupling implemented? Are wave-dependent air-sea momentum fluxes part of the coupling (both to atmosphere and ocean) for instance? It is precisely in situations with rapidly changing weather conditions that these couplings become important (see Rohrs et al., Ocean Dyn., 2012, for estimates of wave dependent fluxes from directional wave rider data and an analysis of upper ocean drift relevant to the paper under review).

AC: The coupling implementation has been described in the manuscript. Page 7, lines 1-6

RC: Although it appears to be a good idea to split the upper ocean drift velocity into "wave", "wind" and "background" components, it really depends on the ability to delineate them, and I'm not entirely convinced in this case. The CARTHE drifters are influenced by the Stokes drift, how much of the LAVA circulation estimates still contain some Stokes drift despite making model based corrections etc.?

AC: The average Stokes drift during the selected hours over which the pre-existing circulation has been estimated using LAVA is 0.02 m/s and 0.04 m/s for the drogued and undrogued drifter cases, respectively. Due to the velocity spreading and averaging performed by LAVA, the uncertainty which arises from the Stokes drift component is small during these time periods. Page 10, lines 21-28

RC: I'm not sure what the authors should do to present a more convincing case, but I would at least need to see a much clearer presentation with more emphasis on the various potential sources of error in the estimates. Alternatives to the LAVA circulation

estimates should definitely be considered (and compared to the LAVA estimates). This work would constitute a major revision of the current manuscript.

AC: In addition, we have added 2 new figures (Figs 6 and 7) to make a more convincing case that the LAVA velocity fields are fair estimates of the pre-existing circulation by plotting the velocity fields on top of 2 consecutive 24-hour averages of SST data from AVISO. The first 24-hour averages for the SST span the time periods over which the high wind analysis windows are defined, where the second 24-hour periods, span the 24-hours following the high wind event. The structures seen in the SST fields line up very well with the LAVA velocity estimates of the pre-existing circulation and the second 24-hour period of SST shows qualitatively how much these structured changed during the high wind event. With these figures, we've hoped to have further convinced the reader that the estimates for the pre-existing regional circulation are adequate for our velocity deconstruction. Descriptions of these figures (6-7) and their implications can be found on: Page 10, lines 12-28 Page 12, lines 9-23 Page 16, lines 17-24

RC: Some minor comments: Eq. (1) isn't an average, "1/h" is lacking.

AC: You are correct. Thank you, the change has been made. Page 3, line 31

RC: A short description of the typical hydrographic structure of the upper ocean in the region (mixed layer depths, buoyancy frequencies, salinity/temperature profiles etc.) would be nice.

AC: A short description as well as a new figure (Fig. 2) has been added to describe the typical stratification and mixed layer depth in terms of temperature, salinity and potential density, in the presence of a typical frontal feature observed during the experiment. Page 7, lines 22-31.

RC: Please add some info about the spectral resolution in the wave model as well. Eq. (2) is not correct for a model with the limit at infinity, there is a cutoff frequency, and presumably a specific shape of the spectrum is assumed.

[Figure]

AC: Proper limits for the wavenumber have been assigned in Eq. (2), and concerns about the spectral resolution have been addressed. Page 6, lines 21-25
* * *
[Figure]

[Figure]

**Fig. 1.** Left: LAVA Drogued Velocity Field. Right: UWIN-CM Ocean Circulation at 0.5 m

[Figure]

[Figure]

**Fig. 2.** Left: LAVA Drogued Velocity Field. Right: UWIN-CM Ocean Circulation at 0.5 m

---

## Author Comment (AC6) · 31 May 2019

RC: General comments: This paper presents very interesting data on near-surface drifter velocities that are interpreted in terms of near surface currents. This interpretation probably requires a clearer explanation of how the drifter velocity related to the surrounding water motion. I strongly encourage the authors to clarify this and resubmit their paper.

AC: We would like to thank Dr. Ardhuin for his constructive review of the current work.

Specific comments: RC: The author write about "surface current" when they actually mean "drifter motion". The distinction is important as the drifter without the drogue will move due to the direct effect of the wind and of the waves, in addition to that of

the current. In Novelli et al. (2017), the undrogued drifter moves at a speed that is significantly larger than the surface current (their figures 10 and 11), of the order of 10 cm/s for 10 m/s wind. However, it is unclear how this difference scales to the open ocean with very different wave ages and vertical mixing: this cannot scale with the Stokes drift (the Stokes drift in the lab is under 0.5% of the wind speed... ). It is very unclear how the difference between water and drifter motion is estimate or corrected.

AC: We agree that it is very difficult to determine how the wave action and especially the wind will affect the movement of the undrogued drifters during the conditions being analyzed here. For this reason, we do not make any correction of the drifter velocity, but choose to leave the manner in which we describe the "surface currents" unchanged. We believe it makes sense to describe the drifter velocities as the average current over the draft depth of the drifter, given the definition portrayed in Eq. (1). We have tried to be more precise in our description of the surface current estimate and the errors associated with the drifter movement due to velocity slip. Page 6, lines 7-10. Page 17, lines 24-26 and lines 10-13

RC: The idea of a "purely wind-driven current" should be clarified, in particular how the time varying wind produces a time-varying current, including a phase shift in time. On page 9, line 9, I guess there is some wind influence already in the "pre-existing regional circulation, $u\_rc$"

AC: Effect of wind and waves on drifters during the low wind period over which the regional circulation estimates have been described in section 3.2, along with new Figures showing SST for validation (Figs. 6-7). Page 10, lines 21-29 Response time of surface currents has also been addressed with supporting literature. Page 11, lines 1-5

RC: There is not a single mention of density, temperature or salinity in the paper. It is expected that the surface response to the wind is very sensitive to the stratification (slippery layers, e.g. Kudryavtsev et al. JPO 1990). So that the present data is impossible to interpret without that information in the context of the wider literature.
AC: A new figure (Fig. 2) has been added to show a typical transects of salinity, temperature, and potential density across a frontal zone in the region measured during the experiment, however not during one of the high wind periods. Fronts like this one were frequently measured during this experiment showing transects very similar to the one shown in Fig. 2. Page 7, lines 22-31. Effects of stratification have also been referenced in Introduction Page 3, lines 10-14.

RC: Numerical models or parameterizations of waves primarily design to get wave heights can disagree a lot on the short wave purely wind-driven current components that contribute to the Stokes drift (e.g. Peureux et al. 2008). Hence it would be good to show a specific model validation on the wave spectrum in the 1 m to 40 m wavelengths regime that dominates the Stokes drift.

AC: We've added a new Figure (Fig. 4) to the manuscript to provide wave validation with available observations from the model during one of the high wind events, showing significant wave height, mean wave direction, wavelength, and mean wave period. With only hourly averages from NDBC buoy 42040 to compare to the model too, we felt this figure was more appropriate than trying to calculate wave spectrum from hourly averaged data. Page 8, lines 11-19

RC: Technical corrections: - Page 1: line 28: replace CARHTE with CARTHE

AC: We have replaced this. Thank you

RC: in the paragraph "Observational data that captures the vertical shear within the first meter of wind-driven surface currents is very limited in the real ocean as well" the authors could reference some important work (Santala & Terray 1992)

AC: We have cited Santala and Terray (1992) Page 2, lines 13-15

RC: Page 2, line 28: "twice as fast as the average current over the first 1m and four times as fast" is misleading as a casual reader could think that over a 2 m/s Gulf Stream he would also have a 2 m/s wind shear. Please give a velocity difference in cm/s and

/ or scale it with the wind speed. Please also note that these shears should be mixed by wave breaking and should thus be much smaller in the open ocean than in the lab or in coastal areas / weak winds. As a result, lab studies are largely irrelevant for the open ocean. In that respect, Sutherland et al. (2016) is a relevant reference.

AC: A reference velocity has been provided in regard to this description of the vertical velocity shear. A note about breaking waves and mixing in regard to decreased velocity shear has also been referenced, citing Sutherland et al. 2016

Page 2, line 24 Page 17, lines 22-26

RC: Page 2: line 31. Classical Ekman theory stricto sensu (in particular the 45âŬę !) does not apply to the real ocean. Please consider at least realistic mixing (Madsen 1977 or Rascle et al. 2009 are better).

AC: Rascle et al. 2009 has been described and cited. Page 3, Lines 13-14

RC: Page 2: line 24: 0.5 m is optimistic

AC: 0.5 m was the value from references listed.

RC: Page 3, line 5: "anywhere from 0.4 % to 5 %" is not a scientific statement. The uncertainty is much less than this range, as most of the variability in horizontally ho-mogeneous conditions is known function of the wind speed and stratification (Ardhuin et al. 2009). Besides, I did not find in Berta et al. a clear number on a "wind-only' component.

AC: This range of findings for the wind-driven current is based off the range of results reported by the different studies cited. The lines were edited for clarity. Berta et al. 2018 states their ageostrophic component, which is dominated by the wind, travels at ∼2 % of the wind speed. Page 3, lines 7-8

RC: Page 4, line 31: please replace "current" with "drifter velocity"

AC: We would like to keep the present convention of referring to the drifter velocities as

measurements of the surface currents themselves, with potential errors described.

RC: Page 5, line 9: The acceleration is not just due to Stokes drift as shown in Novelli et al. (Stokes drift at low wind is under 1% of wind speed).

AC: We were referring to lab experiments where there was no wind influence. This point has been clarified in the text. Page 5, line 22

RC: Page 5: lines 16-19: I would not expect that separation changes so much the mean wind speed in the ocean over the near-surface 10 cm. The radiation stress of the short waves dissipated / reflected by the obect can be relevant, see Longuet-Higgins 1977.

AC: The mechanics of radiation stress on the effect of the drifters is non-trival to diagnose, and is thus beyond the scope of this paper.

RC: Page 5: Given the very different wave age in the lab and in the field, it is not clear at all that the "velocity slip" in the lab can be scaled to the field conditions.

AC: We agree it probably cannot, which is why we choose not to perform a velocity correction to the measured drifter velocities.

RC: Page 6: Please show / give reference to proper validation of wave model in terms of Stokes drift.

AC: Wave model validation provided in Fig. 4 Page 8, lines 11-19

RC: Page 10 line 4-5: please be more specific and replace "wind-driven velocities" by "wind driven drifter velocities"

AC: Again, we would like to keep the present convention of referring to the drifter velocities as measurements of the surface currents themselves, with potential errors described.

RC: Page 12, line 31: "possibly the most significant" is a pretty bold comment given the history of the field (Munk 2002). I would contend that stratification is the elephant

in the room here.

AC: Paragraphs have been edited with the stratification mentioned as well. Page 15, line 15.

RC: Page 13: Please do not use the word "current" unless you are properly explaining how you go from drifter velocity to water velocity. AC: We refer to the drifter velocities as measurements of the surface currents themselves, with potential errors described.

RC: Page 13, line 22: please clarify if that includes the Stokes drift or not. Also, it should be important to discuss the effect of proximity to coast as the wind-driven current are rectified by the shoreline in many datasets. AC: Clarification of the inclusion of stokes drift and effect of shoreline rectification has been made. Page 16, lines 4-9

RC: Page 15, line 1: "The momentum input from large breaking waves into the surface currents" what about rather, "the momentum input and surfing behaviour of undrogued drifters in large breaking waves"

AC: Statement has been changed to include this phenomenon. Page 10-13

RC: Page 15 line 11: " twice as fast " does not make much sense, please provide some scale (wind, Stokes drift ...) you do not expect to go twice as fast in the top meter above a 2 m/s Gulf Stream. Also please discuss stratification.

AC: Absolutely velocity measurements have been provided.

Page 17, lines 22-25

---

## Editor Comment (EC1) · Ilker Fer (Editor) · 20 Jul 2019

Dear John,

Fabrice has sent some comments to me by email (he had waived his anonymity earlier). He apologizes for the delay because of other commitments. It would be great if you could consider these when revising your paper.

Thanks, Ilker

Comments from Fabrice:

General comments: The modifications generally go in the right direction.

On the drifter velocity vs water velocity, there is no way that eq. 1 can be generally valid

in the presence of waves, because the momentum in the wave field is not confined to a layer, but instead connected throughout the water column by the vertical structure of the waves (see also Ardhuin et al. 2008, 2018 for further discussion). I thus insist that the authors distinguish between the drifter velocity and the current velocity. As a result waves drive streaming flows in boundary layers (e.g. Longuet-Higgins 1960) and a very thin film of oil can be accelerated to a much larger velocity than just the average of the Stokes drift across its thickness.

There are other still unclear aspects. The authors state in their reply "We've added a new Figure (Fig. 4) to the manuscript to provide wave validation[ .. ] showing significant wave height, mean wave direction, wavelength, and mean wave period." This misses the point of my question: is the model any good for Stokes drift? That requires taking the wave spectra from the buoy and computing the integral weighted by the frequency cubed.

Page 2 line 24: " the upper 1 m" is only true for frequencies around 16 Hz. Multi-frequency radars, or using other peaks than the main Bragg peaks can provide vertical current shear (Ivonin et al. JGR 2005)

such that vertical shear within this depth is not detectable in the measurement (Stewart and Joy, 1974; Röhrs et al. 2015).

@ARTICLE{Longuet-Higgins1960, author = "Michael S. Longuet-Higgins", title = "Mass transport in the boundary layer at a free oscillating surface", journal = JFM, volume = 8, pages = "293–306", year = 1970, KEYWORDS={mass transport;Stokes drift;viscosity}, }

@ARTICLE{Ardhuin&al.2008b, author = "Fabrice Ardhuin and Alastair D. Jenkins and Kostas Belibassakis", title = "Comments on 'The Three-Dimensional Current and Surface Wave Equations' by {George M}ellor", journal = JPO, volume=38, pages="1340–1349", doi="10.1175/2007JPO3670.1", year = 2008, }

@ARTICLE{Ardhuin&al.2017c, author = "Fabrice Ardhuin and Nobuhiro Suzuki and James C. McWilliams and Nori Aiki", title = "Comments on "A Combined Derivation of the Integrated and Vertically Resolved, Coupled Wave-Current Equations"", journal = JPO, volume=47, number=9, pages="2377–2385", doi="10.1175/JPO-D-17-0065.1", year = 2017, keywords={3D wave-current}, }

---

## Author Response (AR2)

**To the Editor:**

*The major changes made during this revision are as followed:*

- *Change from describing ocean currents to drifter velocities*
- *Wave and calculated stokes drift validation at higher frequencies*
- *Comparison of LAVA velocity fields with alternative sources for regional circulation estimate (NCOM velocities and AVISO geostrophic velocities)*
- *Analysis of the total velocity deconstruction using NCOM velocities and AVISO geostrophic velocities*
- *Sensitivity test involving the subtraction of a theoretical wind-driven current during the low-wind periods.*

**Anonymous Referee #1**

The manuscript has significantly improved with regard to structure and context to a wider range of literature.

*We would like to again, express our gratitude to referee #1 for his or her constructive feedback on the manuscript*

A major issue that has not been addressed in the revised version of the manuscript concerns the identification of a pre-existing ocean circulation at the onset of the wind event (ref. to previous comment).

The authors show that ocean circulation from the LAVA model agrees with SST imagery, but this does not address if any pre-existing wind-driven circulation was present, and how this wind-driven circulation may change with time and interfere with the components that are used during further analysis. This should be addressed in a new version of the paper, or at least those weaknesses and should be clearly stated if they are shown not to affect the results of this study.

*A sensitivity test involving an additional subtraction of a theoretical wind-driven current from the LAVA velocity fields and its effect on the overall results has been presented in the discussion section on pg 18, line 34 – pg 19, line 19.*

*As for how this wind-driven circulation changes in time, we would like to refer the referee to pg 12, lines 16-20, where we cite a study performed by Sentchev et al. (2017) which describes the response time of surface current to changing winds as quasi-instantaneous, with the surface currents lagging the wind by ~40 minutes, which we assume is more or less the case in this study as well.*

*How the wind-driven current will interact and change the regional circulation, visible in the SST data, is a realized shortcoming of the method (described on Pg 19, lines 20-29) used in the manuscript, but it is clearly the best option compared to the results using NCOM velocity fields or geostrophic velocities for the total velocity deconstruction.*

*Additional description, methods, results and discussion of the new data sets used for LAVA comparison can be found on:*

> *Pg 4, lines 10-12, lines 22-23*
> *Pg 7, lines 13-30*
> *Pg 12, lines 1-11*
> *Pg 12, line 31 – Pg 13, line 2*
> *Pg 14, lines 11-18, lines 26-33*
> *Pg 15, lines 3-15*
> *Pg 16, lines 20-31*
> *Pg 17, lines 1-2*

> *At this time, we show the LAVA-derived velocity fields to be the superior option to estimate the regional circulation.*

**Anonymous Referee# 2**

My main concern about the use of LAVA to derive the so-called "pre-existing circulation" has in my opinion not been addressed properly. By all means, LAVA might be a good method to estimate the Eulerian circulation for practical purposes such as oil spill drift modelling, where some uncertainty with regards to the wind and wave induced drift components can be accepted. In this paper, however, where proper identification of the Eulerian background circulation is paramount, I'm still not convinced that it is suitable.

The comparison with SST is inadequate. In some places the LAVA estimates of the currents go across the fronts, in other places along the fronts, and the interpretation will necessarily be subjective. Crude estimates of the circulation from SST can be obtained using maximum cross correlation methods, but even better would be to estimate surface geostrophic velocities from SSH and compare that with the LAVA results (or use the SSH estimates instead). I also wonder why the authors haven't investigated other sources of independent ocean model data that might be more suitable. Numerical models are dismissed out of hand with a couple of references that date back to 2008 and 2011, and I would expect a lot has happened since that time.

I cannot recommend publication unless a proper comparison of LAVA estimates with other estimates are shown, or alternative estimates are used instead for the background circulation, and until that time I'm reluctant to comment on other minor issues in the paper. There is one exception, though:

> *We would like to thank Referee #2 for his or her perspective on the current manuscript.*

> *As requested by the reviewer we have investigated other sources for estimating the regional circulation component used in this study, specifically geostrophic velocities calculated using altimeter based SSH data from AVISO and NCOM ocean circulation*

*velocities that were forecasted during the LASER experiment. Information on the AVISO data and NCOM configuration can be found in the revised paper (pg. 7 lines 14-30).*

*Newly added Figure 9, shows the geostrophic velocities and NCOM velocities, used to perform the same velocity deconstruction performed using the LAVA-derived velocity fields, and plotted on top of corresponding SST from AVISO. Figure 11 now shows results using the NCOM velocity fields for the velocity deconstruction and compares these results to those resulting from the LAVA-derived fields. Reasons for not showing the results associated with the geostrophic velocities are explained in detail in the manuscript, but briefly put, the geostrophic velocities are so small such that there is little change in the full drifter velocities when the geostrophic velocities are subtracted. Results and discussion involving these new data sets can be found on:*

> *Pg 4, lines 10-12, lines 22-23*
> *Pg 7, lines 13-30*
> *Pg 12, lines 1-11*
> *Pg 12, line 31 – Pg 13, line 2*
> *Pg 14, lines 11-18, lines 26-33*
> *Pg 15, lines 3-15*
> *Pg 16, lines 20-31*
> *Pg 17, lines 1-2*

The new Fig. 4 shows that the wave model component consistently underestimates the wave periods. This has a bearing on the accuracy of the Stokes drift estimates. A more in depth comparison of the high frequency part of the wave spectrum, also commenting on the ability of the wave buoy to capture high frequency waves, should be presented.

> *Figure 5 now shows higher frequency wave data and stokes drift calculated from NDBC buoy 42040 and the modeled wave spectrum. Description of the figure can be found on:*

> *Pg 9, lines 6-25*

> *In addition, the authors would like to share two more Figures with the reviewer. The first showing the scatter plots of deflection angles between the drifter velocities and the wind, after NCOM velocities at 2m depth, during the low-wind periods coinciding with the timing of the LAVA-derived velocity fields, are subtracted from the full drifter velocities. 2m depth was chosen to avoid any wind-driven influence in the modeled fields at the surface during these low wind periods. And the second showing the same using the geostrophic velocity which illustrates that there is very little change in the scatter from the full drifter velocity once the geostrophic velocity estimate is removed.*

[Figure]

*Figure 1: Shows increased, disorganized scatter using the NCOM velocities at 2 m depth, during low wind periods for velocity deconstruction.*

[Figure]

*Figure 2: Shows little change to the scatter using the geostrophic velocities for velocity deconstruction.*

**Dr. Fabrice Ardhuin (Referee)**

> *Again, we would like to thank Dr. Ardhuin for his constructive review of the current work.*

General comments:

The modifications generally go in the right direction. On the drifter velocity vs water velocity, there is no way that eq. 1 can be generally valid in the presence of waves, because the momentum in the wave field is not confined to a layer, but instead connected throughout the water column by the vertical structure of the waves (see also Ardhuin et al. 2008, 2018 for further discussion). I thus insist that the authors distinguish between the drifter velocity and the

current velocity. As a result waves drive streaming flows in boundary layers (e.g. Longuet-Higgins 1960) and a very thin film of oil can be accelerated to a much larger velocity than just the average of the Stokes drift across its thickness.

*Eq. 1 has been restated to describe the drifter velocity instead of the average surface current over the given depth of the drifter. Language throughout the manuscript has been changed in accordance.*

There are other still unclear aspects. The authors state in their reply "We've added a new Figure (Fig. 4) to the manuscript to provide wave validation [ .. ] showing significant wave height, mean wave direction, wavelength, and mean wave period." This misses the point of my question: is the model any good for Stokes drift? That requires taking the wave spectra from the buoy and computing the integral weighted by the frequency cubed.

*Another Figure (Fig. 5) showing 2-D wave energy spectrum from the model and NDBC buoy 42040 along with the 1-D wave density spectrum of each as well, along with a stokes drift calculation using modeled and observed wave spectrum, has been added and described:*

*Pg 9, lines 6-25*

Page 2 line 24: " the upper 1 m" is only true for frequencies around 16 Hz. Multi- frequency radars, or using other peaks than the main Bragg peaks can provide vertical current shear (Ivonin et al. JGR 2005)

*This statement has been re-written and citation added to manuscript*

*Pg 2, lines 17-21*

[revised manuscript text omitted]

---

## Author Response (AR3)

The authors have made a serious effort to clarify outstanding questions from the previous reviews, and give a much more complete assessment of the involved uncertainties in their study. Finally, I am convinced that the "pre-existing" ocean circulation does not affect their central analysis to a problematic degree, which is to quantify the response of surface drifters to wind forcing. I would generally recommend publication in OS, after the following questions are addressed:

> *The authors would like to again thank Referee #1 for his or her feedback.*

LAVA circulation estimates are compared with NCOM circulation at 60/15m depth. The authors state that they want to avoid any wind-driven circulation in NCOM, but isn't the pre-existing (weak) wind-driven circulation exactly what could add information in this study?
I would like to see a better clarification why ocean currents at the base of the mixed layer are selected for comparison.

> *The authors use depths at 60m/15m in order to avoid contamination from any instantaneous wind stress incorporated by NCOM during the high wind event. The authors chose to look at modelled velocity fields during the wind event, instead of prior to the wind event, in order to account for any change in the circulation during the increase of winds. We have performed the same analysis, varying depths from 2m to 200m and varying time from the time of the observed LAVA velocities used in this study to the end of the high wind analysis, and results with regard to the scatter plots in Fig. 11, show no significant improvement, showing that the NCOM ocean circulation component is not adequate for our final analysis, no matter the depth or time one chooses to use for analysis.*

> *To the issue of using NCOM to provide information on the pre-existing wind-driven circulation, the NCOM parameterization implemented by the Coupled Ocean/Atmosphere Mesoscale Predictions System (COAMPS) has been shown to underestimate the wind-stress in the upper 1 m (Haza et al., 2019). When predicting CARTHE drifter trajectories with surface velocities from NCOM, Haza et al. (2019) showed that trajectory predictions became more accurate, when an additional velocity input from the wind was applied to the surface currents.*

> *Since the authors don't have a great amount of confidence in the response of the NCOM model to applied wind stress regarding near-surface transport, or the ocean circulation component in the region of interest for this study we don't feel it would be reliable to use these NCOM components to provide estimates of the pre-existing wind-driven circulation.*

> *Revisions related to the clarification of these points can be found on:*
> > *Pg 7 lines 17-27*

> *Please note that a slight error in the text from the previous version of the paper has also*

*been corrected. The NCOM modeled wind stress in forced with winds and parameterization from the Coupled Ocean/Atmosphere Mesoscale Predictions System (COAMPS), not the UWIN-CM atmospheric component. We apologize for any confusion.*
            *Pg 7 lines 18-20*

In the discussion section, the authors conclude that the assimilation of satellite data (SSH and SST) does not help circulation estimate in NCOM in the study region, based on the view that the geostrophic current (AVISO) alone is not a match with current observations.
I strongly disagree, independent on the quality of NCOM circulation estimate during this experiment. The geostrophic balance is only one component in ocean circulation, and I doubt that the geostrophic current vectors are directly assimilated in NCOM. Typically, observed quantities are used in data assimilation, i.e. SSH and SST, and any added variable has the potential to improve the analysis in assimilative models. To quantify the benefit, one needs to perform detailed assimilation impact studies which cannot be done with the methods presented here.

> *The authors do not mean to imply that the NCOM velocities are not improved at all by the data assimilation and we agree that the geostrophic current is only one component in the ocean circulation model, but given the relatively small geostrophic velocities in the region, compared to that of the LAVA-derived and NCOM velocity fields, it seems fair to say that the assimilated altimetry data is only informing the NCOM model to a limited degree based on what is observable from satellite. The region of interest is dominated mainly by features on the order of few 10s of kilometers associated with the Mississippi river outflow, which are small in comparison to the mesoscale altimetry data constraining the model.*

> *If the regional circulation present in this study is not well described by geostrophic balance, then the initial conditions provided by altimetry data is not going to constrain the ocean circulation model in this region, due to a lack of initial conditions that describe the other force balances and scales within the flow.*

> *Paragraph describing this issue has been re-written*
>             *Pg. 16 -17 lines 21- 5*

**Dr. Fabrice Ardhuin (Referee)**

The paper is acceptable as is. I congratulate the authors for the difficult analysis.
Just one minor typo (coming from me) the HF radar frequency mentioned should be 16 MHz and not 16 Hz.

*The authors would like to express gratitude for the constructive feedback throughout the review process.*

*The typo has been fixed.*
        *Pg. 2, line 19*

[revised manuscript text omitted]